



**Composited analyses of the chemical and physical characteristics of co-**
**polluted days by ozone and PM$_{2.5}$ over 2013-2020 in the Beijing–Tianjin–Hebei**
**region**
Huibin Dai[1], Hong Liao[1*], Ke Li[1], Xu Yue[1], Yang Yang[1], Jia Zhu [1], Jianbing Jin[1],
Baojie Li[1]
[1]Jiangsu Key Laboratory of Atmospheric Environment Monitoring and Pollution
Control, Jiangsu Collaborative Innovation Center of Atmospheric Environment and
Equipment Technology, School of Environmental Science and Engineering, Nanjing
University of Information Science & Technology, Nanjing 210044, China
*Correspondence to: Hong Liao (hongliao@nuist.edu.cn)




**Abstract.**
The co-polluted days by ozone ($O_3$) and $PM_{2.5}$ (particulate matter with an
aerodynamic equivalent diameter of 2.5 μm or less) ($O_3\&PM_{2.5}PD$) were frequently
observed in the Beijing–Tianjin–Hebei (BTH) region in warm seasons (April-October)
of 2013-2020. We applied the 3-D global chemical transport model (GEOS-Chem) to
investigate the chemical and physical characteristics of $O_3\&PM_{2.5}PD$ by composited
analyses of such days that were captured by both the observations and the model. Model
results showed that, when $O_3\&PM_{2.5}PD$ occurred, the concentrations of hydroxyl
radical and total oxidant, sulfur oxidation ratio, and nitrogen oxidation ratio were all
high, and the concentrations of sulfate at the surface were the highest among all aerosol
species. We also found unique features in vertical distributions of aerosols during
$O_3\&PM_{2.5}PD$; concentrations of $PM_{2.5}$ decreased with altitude near the surface but
remained stable at 975-819 hPa. Process analyses showed that secondary aerosols
(nitrate, ammonium and sulfate) had strong chemical productions at 913-819 hPa,
which were then transported downward, resulting in the quite uniform vertical profiles
at 975-819 hPa in $O_3\&PM_{2.5}PD$. The weather patterns for $O_3\&PM_{2.5}PD$ were
characterized by a high pressure ridge of the Western Pacific Subtropical High at 850
hPa. The strong southerlies at 850 hPa brought moist air from the south, resulting in a
high RH and hence the strong chemical productions around this layer in $O_3\&PM_{2.5}PD$.
The physical and chemical characteristics of $O_3\&PM_{2.5}PD$ are quite different from
those of polluted days by either $O_3$ alone or $PM_{2.5}$ alone, which have important
implications for air quality management.
Keywords: Co-occurrence, Ozone and $PM_{2.5}$, Pollution, Meteorological parameters.



## 1. Introduction

Surface ozone ($O_3$) and $PM_{2.5}$ (particulate matter with an aerodynamic equivalent diameter of 2.5 micrometers or less) are important air pollutants in the atmosphere that have harmful effects on public health (Gao and Ji, 2018; Jiang et al., 2019), ecosystems (Ren et al., 2011; Yue et al., 2017), and crops (Wang et al., 2005; Wang et al., 2007). Surface $O_3$ is a secondary pollutant produced by photochemical oxidation of volatile organic compounds (VOCs) and nitrogen oxides ($NO_x \equiv NO+NO_2$) in the presence of intense ultraviolet light, and the major $PM_{2.5}$ components (nitrate ($NO_3^-$), ammonium ($NH_4^+$), sulfate ($SO_4^{2-}$), black carbon (BC), organic carbon (OC)) are caused by anthropogenic emissions of aerosols and aerosol precursors. Although surface $O_3$ and $PM_{2.5}$ have different formation mechanisms, they are coupled through the common precursors ($NO_x$ and VOCs) and photochemical reactions (Chu et al., 2020). Since 2013, stringent clean air actions have been implemented to improve air quality in China (State Council of the People's Republic of China, 2013, 2018). However, $O_3$ concentrations increased unexpectedly, while $PM_{2.5}$ concentrations decreased drastically in the past years (Li et al., 2019). The co-polluted days by $O_3$ and $PM_{2.5}$ (concentrations of both $O_3$ and $PM_{2.5}$ exceed the national air quality standards on the same day, hereafter referred to as $O_3\&PM_{2.5}PD$) were also reported (Dai et al., 2019). Therefore, it is fundamental to examine the chemical and physical characteristics of $O_3\&PM_{2.5}PD$.

The Beijing–Tianjin–Hebei (BTH) region is the most populated region in northern China. In the past few years, concentrations of $O_3$ and $PM_{2.5}$ in the BTH were among the highest in China. The Observations from China National Environmental Monitoring Center (CNEMC) showed that the mean and maximum MDA8 (daily maximum 8-h average) $O_3$ in North China in summer of 2019 were 83 ppb and 129 ppb, respectively, and the summer mean MDA8 $O_3$ increased with a trend of 3.3 ppb $a^{-1}$ over 2013–2019





(Li et al., 2020). Gong et al. (2020) reported that $O_3$ polluted days (i.e., MDA8 $O_3$
concentration exceeds 80 ppb) in May-July in the BTH increased from 35 days in the
year of 2014 to 56 days in 2018. As for observed $PM_{2.5}$, the concentration averaged
over BTH had a decreasing trend of 10 $\mu g\,m^{-3}\,yr^{-1}$ over 2013-2019, and the mean value
was $79 \pm 17\,\mu g\,m^{-3}$ over these years (Li et al., 2020). BTH also had the highest
frequency and intensity of severe haze pollution days (i.e., days with daily mean $PM_{2.5}$
concentration exceeding 150 $\mu g\,m^{-3}$) in China from 2013 to 2017, with an observed
mean frequency of 21.2 d $yr^{-1}$ and an observed mean intensity of 231.6 $\mu g\,m^{-3}$ (Dang
and Liao, 2019).
The interactions between $O_3$ and $PM_{2.5}$ have been reported in previous studies.
Zhu et al. (2019) examined the spatial-temporal characteristics of the correlations
between observed $O_3$ and $PM_{2.5}$ at 1497 sites in China for 2016 and found that $O_3$–
$PM_{2.5}$ had the highest positive correlations (correlation coefficients > +0.7) in July in
southern China and the largest negative correlations (r values < −0.5) during January in
northern China. Li et al. (2019) used the GEOS-Chem model to analyze the $O_3$-$PM_{2.5}$
relationship in northern China and found that $O_3$ production was suppressed under high
$PM_{2.5}$ conditions ($PM_{2.5}$ concentrations > 60 $\mu g\,m^{-3}$) because of the reactive uptake of
hydrogen oxide radicals ($HO_x$) by aerosol particles. Chu et al. (2020) analyzed the
observed daily $PM_{2.5}$ and $O_3$ concentrations in 114 cities in China during years of 2013-
2018 and found that the correlations between $O_3$ and $PM_{2.5}$ tended to change from
negative in 2013 to positive in 2018 in China as air quality improved.
Few previous studies have examined the co-occurrence of $O_3$ and $PM_{2.5}$ pollution
(MDA8 $O_3$ > 80 ppb and $PM_{2.5}$ > 75 $\mu g\,m^{-3}$). Zong et al. (2021) used the obliquely
rotated principal component analysis in the T-mode (T-PCA) method to identify the
synoptic weather pattern associated with $O_3\&PM_{2.5}PD$ in eastern China during





summer of 2015–2018, and found that $O_3$&$PM_{2.5}$PD were associated with a stable
western Pacific subtropical high ridge, which brought warm and moist air flow from
the East China Sea to the eastern China to promote hygroscopic growth of fine
particulate matter in BTH and northern YRD. Dai et al. (2021) analyzed $O_3$&$PM_{2.5}$PD
in the YRD for April-October of 2013-2019 by using observations and reported that
the co-polluted days occurred mainly in April (29.6% of co-polluted days occurred in
April), May (23.0%), June (19.5%), and October (10.8%) under meteorological
conditions of higher relative humidity, higher surface air temperature, and lower wind
speed relative to the days with $O_3$ pollution alone. Qin et al. (2021) investigated
$O_3$&$PM_{2.5}$PD by using the hourly observed concentrations of water-soluble ions, OC,
and elemental carbon (EC) in 2019 in cities of Nanjing and Changzhou. They found
that inorganic aerosols mainly existed as $NH_4NO_3$ and the correlation coefficients
between the secondary components $NO_3^-$, $NH_4^+$, and $SO_4^{2-}$ were relatively high during
$O_3$&$PM_{2.5}$PD in 2019, indicating a significant formation of secondary inorganic
aerosols. Although these studies have discussed the meteorological conditions and
some chemical characteristics of $O_3$&$PM_{2.5}$PD, the understanding of $O_3$&$PM_{2.5}$PD
was quite limited because of the limited observations of chemical species involved.

In this work, we take advantage of the comprehensive chemical mechanism of

the global chemical transport model to have better understanding of $O_3$&$PM_{2.5}$PD. We
apply the 3-D global chemical transport model (GEOS-Chem) to simulate
$O_3$&$PM_{2.5}$PD in BTH in years of 2013-2020, and investigate the chemical and
physical characteristics of $O_3$&$PM_{2.5}$PD by composited analyses of such days that are
captured by both the observations and the model. The objectives of this study are: 1)
to examine the underlying chemical mechanisms for $O_3$&$PM_{2.5}$PD in BTH for warm
seasons (April-October) of 2013-2020 by comparing $O_3$&$PM_{2.5}$PD with polluted days





by $O_3$ alone or by $PM_{2.5}$ alone, and 2) to identify the weather patterns that are
associated with $O_3$&$PM_{2.5}$PD in BTH. The observations, the reanalyzed
meteorological data, the GEOS-Chem model, and the process analysis are described
in Section 2. The observed $O_3$&$PM_{2.5}$PD are presented in Section 3.1. The evaluation
of simulated concentrations of $O_3$ and $PM_{2.5}$ as well as the simulated pollution days by
$O_3$ and/or $PM_{2.5}$ are shown in Section 3.2. The underlying mechanisms of
$O_3$&$PM_{2.5}$PD are shown in Section 3.3. In Section 3.4, the meteorological conditions
for the co-occurrence of $O_3$ and $PM_{2.5}$ pollution are investigated. The conclusions are
presented in Section 4.

**2. Methods**
**2.1 Observed $O_3$ and $PM_{2.5}$ concentrations**
Hourly concentrations of $PM_{2.5}$ and $O_3$ in China over the years of 2013-2020
were taken from the public website of CNEMC (https://air.cnemc.cn:18007/,
CNEMC, 2022). To ensure data quality, the daily mean $PM_{2.5}$ concentration was
calculated when there were valid data for more than 20 h during that day and the
MDA8 $O_3$ concentration was calculated when there were valid data for at least 6 h for
each 8 h. For the calculation of monthly and annual mean concentrations, the number
of days with valid concentrations had to be more than 15 in each month. The spatial
distribution of the 79 valid sites within BTH (37-41°N, 114-118°E, the black
rectangle) is shown in Fig. 1. For model evaluation, the observed concentrations were
averaged over sites within each of the 0.5° latitude × 0.625° longitude MERRA-2
grid cell. There are 18 model grids in BTH. Note that the observed $O_3$ concentrations
from this network have a unit of $\mu g\ m^{-3}$. For the consistency of observed and
simulated $O_3$ concentrations, 1 $\mu g\ m^{-3}$ of $O_3$ is approximately 0.5 ppb under the

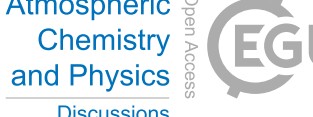

conditions of 298 K and 1013 hPa. The observed $O_3$ concentrations reported by the
CNEMC were under standard conditions of 273 K and 1013 hPa before 31 August
2018 and were under standard conditions of 298 K and 1013 hPa afterwards
(http://www.mee.gov.cn/ xxgk2018/xxgk/xxgk01/201808/t20180815_629602.html),
which were accounted for as $O_3$ concentrations were converted to ppb.
According to the National Ambient Air Quality Standard of China (GB3095-
2012), $O_3$ ($PM_{2.5}$) concentration exceeds the national air quality standard if the MDA8
$O_3$ (daily mean $PM_{2.5}$) concentration is higher than 160 μg m$^{-3}$ (75 μg m$^{-3}$). In this
study, we define $O_3$ polluted days (hereafter called '$O_3$PD') for days with MDA8 $O_3$
concentration > 160 μg m$^{-3}$, $PM_{2.5}$ polluted days (hereafter called '$PM_{2.5}$PD') with
daily mean $PM_{2.5}$ concentration > 75 μg m$^{-3}$, and the co-pollution days by $O_3$ and
$PM_{2.5}$ ($O_3$&$PM_{2.5}$PD) with daily MDA8 $O_3$ concentration > 160 μg m$^{-3}$ as well as the
daily mean $PM_{2.5}$ concentration > 75 μg m$^{-3}$.

**2.2 Reanalyzed meteorological fields**
Meteorological fields were obtained from the Version 2 of Modern Era
Retrospective-analysis for Research and Application (MERRA2), which were
generated by the NASA Global Modeling and Assimilation Office (GMAO). The
MERRA2 data have a horizontal resolution of 0.5° latitude × 0.625° longitude and
72 vertical layers (Molod et al., 2015). To analyze the meteorological conditions for
$O_3$&$PM_{2.5}$PD, vertical pressure velocity (OMEGA), planetary boundary layer height
(PBLH), temperature (T), relative humidity (RH), surface incoming shortwave flux
(SWGDN) are used. Note that the temporal resolution for PBLH, T, and SWGDN is
1h, and that for OMEGA and RH is 3h. Daily mean geopotential heights at 850 and
500 hPa from the National Center for Environmental Prediction (NCEP) and National





Center for Atmospheric Research (NCAR) global reanalysis with a resolution of 2.5°
latitude by 2.5° longitude are also utilized in this study.

**2.3 Observed aerosol optical depth**

We obtained the version 3 datasets of observed daily aerosol optical depth

(AOD) of level 2 (improved cloud screened and quality-assured) from Aerosol
Robotic Network (AERONET, https://aeronet.gsfc.nasa.gov/new_web/index.html)
established by NASA and LOA-PHOTONS (Giles et al., 2019). Three sites in the
BTH region have observations available over 2013-2020, including Beijing (39.97°N,
116.38°E), Beijing-CAMS (39.93°N, 116.31°E), and Xianghe (39.75°N, 116.96°E).
The AOD values at 440 nm and 675 nm at these three sites are analyzed in this study.

**2.4 GEOS-Chem model**

We simulated $O_3$ and $PM_{2.5}$ using the nested version of the 3-D global chemical

transport model (GEOS-Chem, version 11-01) driven by the MERRA2
meteorological data. The nested domain was set over Asia (60°-150°E,11°S-55°N)
with a horizontal resolution of 0.5° latitude × 0.625° longitude, and the chemical
boundary conditions were provided by the global GEOS-Chem simulation with 2.5°
latitude × 2.5° longitude horizontal resolution.

The GEOS-Chem model includes fully coupled $O_3$-$NO_x$-hydrocarbon and

aerosol chemistry mechanism (Bey et al., 2001; Pye et al., 2009) to simulate aerosols
including $SO_4^{2-}$ (Park et al., 2004), $NO_3^-$ (Pye et al., 2009), $NH_4^+$, BC and OC (Park
et al., 2003), mineral dust (Fairlie et al., 2007), and sea salt (Alexander et al., 2005) as
well as the gas-phase pollutants such as $NO_x$ and $O_3$. Over the Asian domain, the



anthropogenic emissions of OC, BC, carbon monoxide (CO), sulfur dioxide ($SO_2$),
$NO_x$, ammonia ($NH_3$), and VOCs were obtained from the Multi-resolution Emission
Inventory for China (MEIC), which includes emissions from industry, power,
residential and transportation sectors for years of 2014-2017 (Li et al., 2017; Zheng et
al., 2018), 2019 and 2020 (Zheng et al., 2021). Emissions in 2018 were obtained by
the interpolation of those in 2017 and 2019 for each grid due to the lack of publicly
accessible emission inventories for that year. The biogenic emissions in GEOS-Chem
are simulated using MEGAN v2.1 (Guenther et al., 2012).
The hourly $O_3$ and $PM_{2.5}$ concentrations for the years of 2013-2020 were
simulated by the GEOS-Chem model which were driven by MERRA-2
meteorological fields. The model was spinned up for 6 months before the integration
over the studied time period.

**2.5 Process analysis**

Process analysis (PA) was applied to identify the relative importance of
atmospheric processes in $O_3$&$PM_{2.5}$PD. PA has been widely used in previous studies
to examine the key processes contributing to air pollution episodes (Gonçalves et al.,
2009; Dang and Liao, 2019; Gong and Liao, 2019) as well as the interannual and
decadal variations of air pollutants (Mu and Liao, 2014; Lou et al., 2015). Five major
processes that influence $O_3$ and $PM_{2.5}$ concentrations were diagnosed at every time
step, including net chemical production, dry deposition, horizontal advection, vertical
advection, and diffusion, for the regional pollution days (days with more than half of
the sites in BTH experiencing pollutions). We carried out PA for $O_3$SPD (exclude
$O_3$&$PM_{2.5}$PD from $O_3$PD), $PM_{2.5}$SPD (exclude $O_3$&$PM_{2.5}$PD from $PM_{2.5}$PD), and
$O_3$&$PM_{2.5}$PD over BTH.




## 3. Results

### 3.1 Observed polluted days by $O_3$ and $PM_{2.5}$

Figure 1a shows the spatial distributions of observed numbers of $O_3PD$, $PM_{2.5}PD$, and $O_3\&PM_{2.5}PD$ summed over the warm seasons (April-October) of 2013-2020. The spatial distributions of polluted days in each year are shown in Fig. S1. The numbers of $O_3PD$, $PM_{2.5}PD$, and $O_3\&PM_{2.5}PD$ were high in BTH, which were, respectively, 524.3 344.6, and 128.1 days from observations, as the values were averaged over all sites in BTH. The high numbers of $O_3PD$, $PM_{2.5}PD$, and $O_3\&PM_{2.5}PD$ in BTH were associated with the highest anthropogenic emissions ($NO_x$ and NMVOCs) in this region (Dang et al., 2021).

Figure 1b shows the numbers of days averaged over all sites in BTH for non-polluted days (NPD, MDA8 $O_3$ < 80 ppb and $PM_{2.5}$ < 75 µg m$^{-3}$), $O_3PD$, $O_3\&PM_{2.5}PD$, and $PM_{2.5}PD$ in each month of warm seasons from 2013 to 2020. $O_3PD$ and $O_3\&PM_{2.5}PD$ mainly occurred in May, June, and July, while $PM_{2.5}PD$ mainly appeared in April and October. The monthly numbers of $O_3\&PM_{2.5}PD$ ($PM_{2.5}PD$) declined from 2013 to 2020, with the fastest drop in June, from 7.5 (17.1) days in June 2013 to 1.8 (1.8) days in June 2020. On the contrary, the numbers of $O_3PD$ kept increasing, especially in June, from 10.9 days in June 2013 to 23.6 days in June 2020. The reductions in $O_3\&PM_{2.5}PD$ were associated with the large reductions in $PM_{2.5}$ since the implementation of the Clean Air Action in 2013.



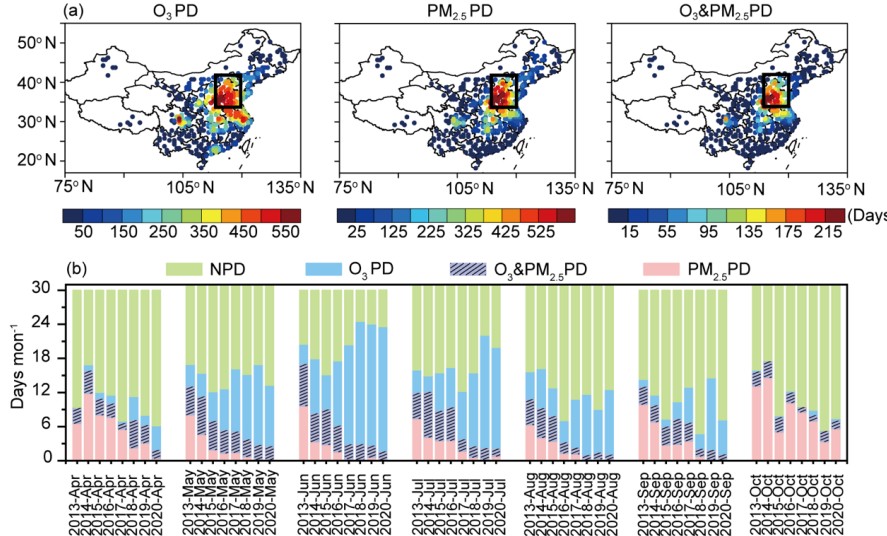

**Figure 1.** (a) Spatial distributions of observed numbers of $O_3PD$, $PM_{2.5}PD$, and

$O_3\&PM_{2.5}PD$ summed over April-October of 2013-2020. The solid black rectangle

indicates the BTH region. (b) The observed numbers of NPD (non-polluted days,

green), $O_3PD$ (blue + purple with slashes), $O_3\&PM_{2.5}PD$ (purple with slashes), and

$PM_{2.5}PD$ (pink + purple with slashes) averaged over all sites in BTH from April to

October in 2013 to 2020.

Figure 2a shows the linear trends of observed $O_3PD$, $PM_{2.5}PD$, and $O_3\&PM_{2.5}PD$

in warm seasons of 2013-2020 averaged over the BTH. $O_3PD$ showed an upward

trend of 7.9 days $yr^{-1}$ from 2013 to 2020. However, the numbers of $PM_{2.5}PD$ and

$O_3\&PM_{2.5}PD$ decreased over 2013-2020, with linear trends of -11.2 and -3.4 days $yr^{-1}$,

respectively. Figure 2b shows the changes in percentage of $O_3\&PM_{2.5}PD$ in $PM_{2.5}PD$

from 2013 to 2020 for each month. It should be noted that, when $PM_{2.5}PD$ occurred,

the proportions of $O_3\&PM_{2.5}PD$ had an upward trend from 2013 to 2020. In May,

June, August, and September of 2020, the proportions of $O_3\&PM_{2.5}PD$ in $PM_{2.5}PD$

reached 100%, indicating that $PM_{2.5}$ pollution was accompanied by $O_3$ pollution in

recent years.

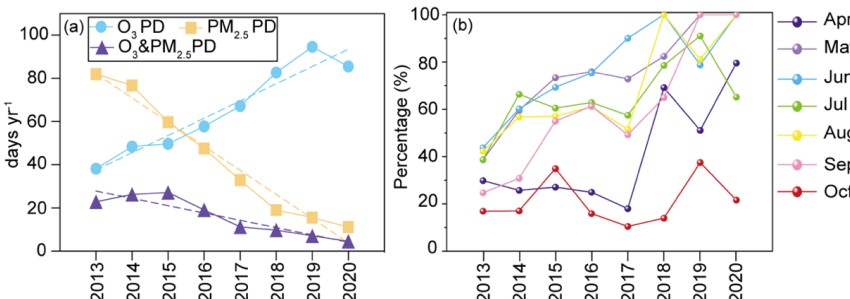


**Figure 2.** (a) The trends of observed $O_3$PD, $PM_{2.5}$PD, and $O_3$&$PM_{2.5}$PD in warm

seasons from 2013 to 2020 averaged over all sites in BTH. The blue, yellow and purple
solid lines (dashed lines) represent the numbers (liner trend) of $O_3$PD, $PM_{2.5}$PD, and
$O_3$&$PM_{2.5}$PD, respectively. (b) The percentage of $O_3$&$PM_{2.5}$PD in $PM_{2.5}$PD for April
to October in 2013 to 2020. The polluted days were averaged over all sites in BTH.

**3.2 Simulated polluted days and model evaluation**
**3.2.1 Simulated surface-layer MDA8 $O_3$ and $PM_{2.5}$ concentrations**
Figures 3a and 3b show, respectively, the spatial distributions of simulated and
observed surface-layer concentrations of MDA8 $O_3$ and $PM_{2.5}$ in China, as the
concentrations are averaged over the warm seasons (April-October) of 2013-2020.
The concentrations of MDA8 $O_3$ and $PM_{2.5}$ were both high in BTH. Averaged over
BTH and the studied time period, the observed concentrations of MDA8 $O_3$ and $PM_{2.5}$
were 58.1 ppb and 60.3 μg m$^{-3}$, respectively, while the simulated values were 68.0 ppb
and 61.1 μg m$^{-3}$, respectively. Figures 3c and 3d compare the time series of observed
and simulated daily MDA8 $O_3$ and $PM_{2.5}$ concentrations averaged over the BTH. The
simulated daily concentrations of MDA8 $O_3$ ($PM_{2.5}$) in eight warm seasons have a
normalized mean bias (NMB) of 7.9% (10.6%). The model generally captures the
daily variations (peaks and troughs) in the observed MDA8 $O_3$ and $PM_{2.5}$



concentrations, with R values of 0.80 and 0.72, respectively.

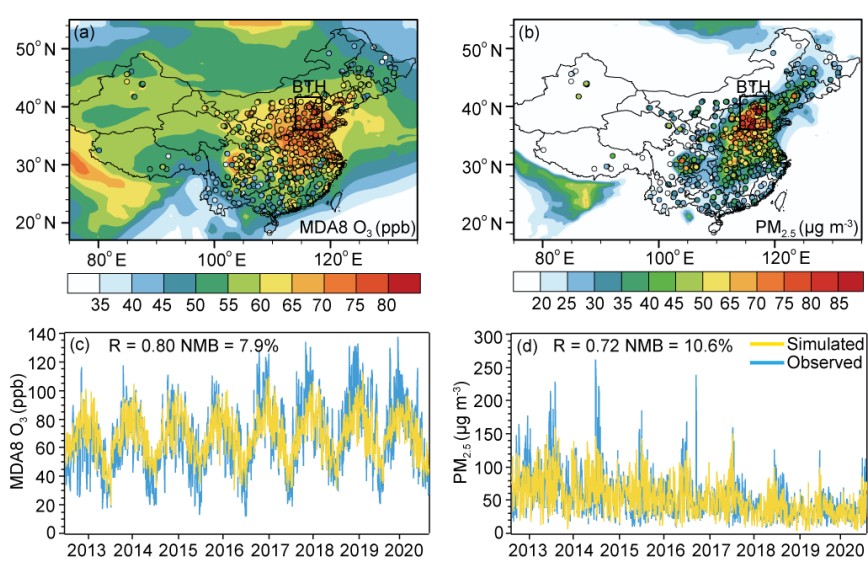


**Figure 3.** Spatial distributions of simulated (shades) and observed (CNEMC, dots)
surface-layer concentrations of (a) MDA8 $O_3$ (ppb) and (b) $PM_{2.5}$ (µg m$^{-3}$) averaged
over the eight warm seasons (April to October, 2013–2020). The solid black rectangle
in (a) and (b) indicates the BTH region. Simulated and observed daily concentrations
of surface-layer (c) MDA8 $O_3$ and (d) $PM_{2.5}$ averaged over BTH. The correlation
coefficient (R) and normalized mean bias (NMB) are also shown for (c) and (d).
NMB $= (\sum_{i=1}^{N}(M_i - O_i)/\sum_{i=1}^{N}(O_i)) \times 100\%$, where $O_i$ and $M_i$ are the observed and
simulated concentrations, respectively, $i$ refers to the $i^{th}$ day, and $N$ is the total number
of days.
**3.2.2 Simulated $O_3$PD, $PM_{2.5}$PD, and $O_3$&$PM_{2.5}$PD**

Figure S2 shows the capability of the model in capturing the polluted days.

Although the GEOS-Chem model well reproduces the spatial distributions of observed
MDA8 $O_3$ and $PM_{2.5}$ concentrations, it underestimates the numbers of $O_3$PD, $PM_{2.5}$PD,
and $O_3$&$PM_{2.5}$PD because of the model's deficiency in capturing the peak



concentrations of air pollutants. Such deficiency was also reported in previous studies
that used the GEOS-Chem model or the weather Research and Forecasting with
Chemistry (WFR-chem) model (Zhang et al., 2016; Ni et al., 2018; Gong and Liao,
2019; Dang and Liao, 2019). Therefore, to identify $O_3PD$, $PM_{2.5}PD$, and $O_3\&PM_{2.5}PD$
using model results, we utilized lower thresholds by considering the NMBs of simulated
MDA8 $O_3$ and $PM_{2.5}$ concentrations in each of 18 grids of BTH. Taking the grid of
Beijing as an example, simulated MDA8 $O_3$ and $PM_{2.5}$ had NMBs of -22.0% and -
26.9%, respectively, as the simulated concentrations were compared with observations
for days with observed concentrations higher than the national air quality standards over
the warm seasons of 2013-2020. We then adjusted the threshold of $O_3PD$ in this grid to
be 62.4 ppb (80 ppb×78%) and that of $PM_{2.5}PD$ to be 54.8 μg m$^{-3}$ (75 μg m$^{-3}$×73.1%).
These adjusted thresholds were also used to identify $O_3\&PM_{2.5}PD$. Such approach was
also used in previous studies to better capture the pollution events based on the
simulations (Dang and Liao, 2019; Gong and Liao, 2019). With the adjusted thresholds,
56-93% of the observed $O_3PD$, $PM_{2.5}PD$, and $O_3\&PM_{2.5}PD$ can be captured by the
model (Fig. S2e).
**3.2.3 Simulated $O_3SPD$, $PM_{2.5}SPD$, and $O_3\&PM_{2.5}PD$**
Since $O_3PD$ or $PM_{2.5}PD$ encompasses $O_3\&PM_{2.5}PD$, we further define $O_3$ single
pollution days (hereafter called "$O_3SPD$", which is to exclude $O_3\&PM_{2.5}PD$ from $O_3PD$)
and $PM_{2.5}$ single pollution days (hereafter called "$PM_{2.5}SPD$", which is to exclude
$O_3\&PM_{2.5}PD$ from $PM_{2.5}PD$) for the purpose of obtaining the characteristics of
different polluted days. Figures 4a and 4b show, respectively, the spatial distributions
of numbers of $O_3SPD$, $PM_{2.5}SPD$, and $O_3\&PM_{2.5}PD$ from observations and from the
GEOS-Chem model using the adjusted thresholds. Considering the total of polluted
days in 18 grids in BTH, observed $O_3SPD$, $PM_{2.5}SPD$, and $O_3\&PM_{2.5}PD$ were,



respectively, 3937, 3698, and 2024 days, in which 75.0% (2954/3937), 58.1%
(2148/3698), and 79.7% (1614/2024) were captured by observation and simulation
simultaneously (Fig. 4c). In addition, the numbers of observed and captured $O_3$SPD,
$PM_{2.5}$SPD, and $O_3$&$PM_{2.5}$PD in each month are shown in Fig. S3. The model has a
fairly good capability of capturing the observed polluted days in each month.

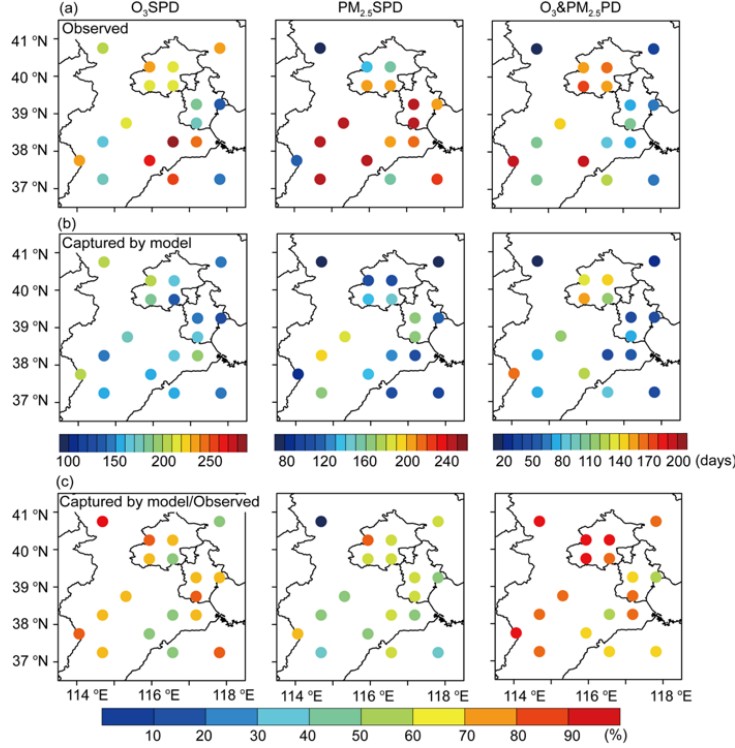


**Figure 4.** Spatial distributions of (a) observed numbers of $O_3$SPD, $PM_{2.5}$SPD, and
$O_3$&$PM_{2.5}$PD, (b) numbers of polluted days that were observed and also captured by
the GEOS-Chem model with adjusted thresholds, and (c) percentages of observed
polluted days that were captured by the model with adjusted thresholds. The values
were calculated for the warm months (April to October) of 2013-2020.

**3.3 Chemical characteristics of polluted days by $O_3$ and $PM_{2.5}$**



### 3.3.1 Atmospheric oxidants of O₃SPD, PM₂.₅SPD, and O₃&PM₂.₅PD

Figure 5 shows the boxplots of daily concentrations of hydroxyl radical (OH) and total oxidant ($O_x = O_3 + NO_2$) from the model for days of O₃SPD, PM₂.₅SPD, and O₃&PM₂.₅PD that were observed and also captured by the model (samples in Fig. 4b) in the warm seasons of 2013-2020 in 18 grids of BTH. The levels of OH and $O_x$ characterize the atmospheric oxidation capacity, following Hu et al. (2020) and Chan et al. (2017). The concentrations of OH were the highest in O₃SPD, with an averaged value of $2.8 \times 10^6$ molec cm$^{-3}$, followed by that in O₃&PM₂.₅PD ($2.0 \times 10^6$ molec cm$^{-3}$) and in PM₂.₅SPD ($1.0 \times 10^6$ molec cm$^{-3}$). Due to the lack of publicly accessible observations of OH in BTH, we compare the simulated OH concentrations with observations reported in the literature. The simulated OH concentrations agree closely with the observed values. In Wangdu of BTH, while the observed daily maximum OH concentrations in summer of 2014 were in the range of $5\text{-}15 \times 10^6$ molec cm$^{-3}$ (Tan et al., 2016), the simulated OH concentrations in the same time period in this work were 3.7-$9.5 \times 10^6$ molec cm$^{-3}$. In Beijing in summer of 2017, the observed daily mean OH concentration was $5.8 \times 10^6$ molec cm$^{-3}$ (Woodward et al., 2020) and our simulated value was $2.4 \times 10^6$ molec cm$^{-3}$.

The mean values of $O_x$ were, respectively, 178.7, 118.1, and 184.1 μg m$^{-3}$ in O₃SPD, PM₂.₅SPD, and O₃&PM₂.₅PD, indicating that the atmospheric oxidation capacity was strong in O₃&PM₂.₅PD, which favored the production of secondary components of PM₂.₅. Figure 5 also shows sulfur oxidation ratio (SOR, n-$SO_4^{2-}$ / (n-$SO_4^{2-}$ + n-SO₂), where n-$SO_4^{2-}$ and n-SO₂ are the concentrations of $SO_4^{2-}$ and SO₂, respectively) and nitrogen oxidation ratio (NOR, n-$NO_3^-$ / (n-$NO_3^-$ + n-NO₂), where n-$NO_3^-$ and n-NO₂ are the concentrations of $NO_3^-$ and NO₂, respectively). SOR and NOR are measures of the conversion degrees of sulfur and nitrogen, respectively (Zhu et al.,





2019). In $O_3$SPD, $PM_{2.5}$SPD, and $O_3$&$PM_{2.5}$PD, the averaged values of SOR were
50.0%, 36.7%, and 49.7%, and those of NOR were 55.4%, 70.0%, and 70.2%,
respectively. The high SOR and NOR in $O_3$&$PM_{2.5}$PD indicated the strong formation
of $SO_4^{2-}$ and $NO_3^-$ that were promoted by high atmospheric oxidation capacity. The
monthly variations of OH, $O_x$, and SOR were similar (Fig. 5), with the highest values
in summer, owing to the high temperature that promoted high concentrations of
oxidants and SOR. It is interesting that SOR and $O_x$ values were higher in $O_3$&$PM_{2.5}$PD
than in $O_3$SPD or in $PM_{2.5}$SPD during May-August. Similarly, NOR values were higher
in $O_3$&$PM_{2.5}$PD than in $O_3$SPD or in $PM_{2.5}$SPD in May and July-September. Overall,
the $O_3$&$PM_{2.5}$PD occurred with high levels of atmospheric oxidants, SOR, and NOR,
leading to combined increases in $O_3$ and $PM_{2.5}$ concentrations.

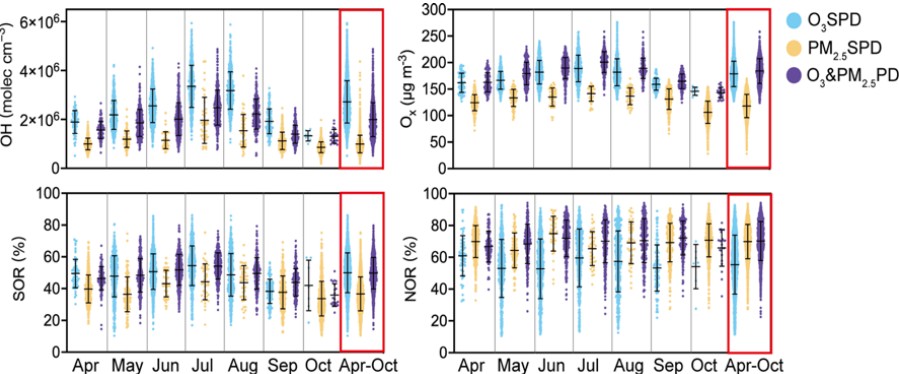


**Figure 5.** The boxplots of surface-layer hydroxyl radical (OH, molec cm$^{-3}$), total
oxidant ($O_x$, μg m$^{-3}$), sulfur oxidation ratio (SOR, %), nitrogen oxidation ratio
(NOR, %) for model-captured $O_3$SPD, $PM_{2.5}$SPD, and $O_3$&$PM_{2.5}$PD in 18 grids of
BTH in the months of April to October from 2013 to 2020. The whiskers represent the
standard deviation, the black line represents the mean value of the samples.
**3.3.2 Surface-layer concentrations of $PM_{2.5}$ components in $O_3$SPD, $PM_{2.5}$SPD, and**
**$O_3$&$PM_{2.5}$PD**

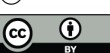

The simulated concentrations of PM$_{2.5}$ components (NO$_3^-$, NH$_4^+$, SO$_4^{2-}$, BC, and
OC, averaged over 18 grids of BTH are shown in Fig. 6 for days of O$_3$SPD, PM$_{2.5}$SPD,
and O$_3$&PM$_{2.5}$PD in the warm seasons of 2013-2020 that were observed and also
captured by the model. While the mean concentrations of NO$_3^-$, NH$_4^+$, BC, and OC,
were all the highest in PM$_{2.5}$SPD, SO$_4^{2-}$ concentration was the highest in O$_3$&PM$_{2.5}$PD.
In O$_3$SPD, PM$_{2.5}$SPD, and O$_3$&PM$_{2.5}$PD, the mean concentrations of SO$_4^{2-}$ were 6.2,
9.4, and 11.97 µg m$^{-3}$, respectively, and the percentages of SO$_4^{2-}$ in PM$_{2.5}$ were 14.9%,
9.0%, and 15.0%, respectively. In July and August, the concentrations of SO$_4^{2-}$ and
MDA8 O$_3$ in O$_3$&PM$_{2.5}$PD were the highest compared with those in O$_3$SPD and
PM$_{2.5}$SPD (Fig. S4).

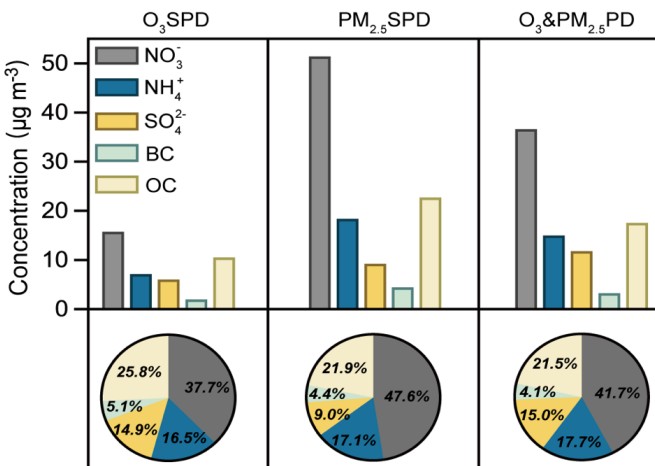


**Figure 6.** The concentrations of PM$_{2.5}$ components (µg m$^{-3}$) and percentages of PM$_{2.5}$
components (%) at the surface for NO$_3^-$, NH$_4^+$, SO$_4^{2-}$, BC, and OC. The values were
averaged over the model-captured O$_3$SPD, PM$_{2.5}$SPD, and O$_3$&PM$_{2.5}$PD in the months
of April to October of 2013-2020 in BTH.
Figure 7 presents the hourly concentrations of NO$_3^-$, NH$_4^+$, SO$_4^{2-}$, BC, OC, and O$_3$
for model-captured O$_3$SPD, PM$_{2.5}$SPD, and O$_3$&PM$_{2.5}$PD over all 18 grids of BTH in





the warm seasons from 2013-2020. Concentrations of $NO_3^-$ and $NH_4^+$ had similarities
in diurnal variations, all of which reached the highest values in the early morning (5:00
local time (LT) in $O_3$SPD and $O_3$&$PM_{2.5}$PD, 7:00-8:00 LT in $PM_{2.5}$SPD) and had the
lowest values in the late afternoon (18:00 LT in $O_3$SPD and $O_3$&$PM_{2.5}$PD, 16:00 LT in
$PM_{2.5}$SPD). Concentrations of BC and OC peaked at the same time as those of $NO_3^-$
and $NH_4^+$ and had the lowest values at 15:00 LT in $O_3$SPD, $PM_{2.5}$SPD, and
$O_3$&$PM_{2.5}$PD. The diurnal variations in $NO_3^-$, $NH_4^+$, BC, OC reflected the diurnal
variations in PBLH (shown in Fig. S5), which generally reached their highest
concentrations before the sudden uplift of PBLH in the early morning (times for uplift
of PBLH: 6:00 LT in $O_3$SPD and $O_3$&$PM_{2.5}$PD, 7:00 LT in $PM_{2.5}$SPD ). Compared to
$O_3$SPD and $O_3$&$PM_{2.5}$PD, the PBLH of $PM_{2.5}$SPD was lower and uplifted one hour
later, which was more favorable for the accumulation of aerosols. During the daytime,
PBLH in $O_3$&$PM_{2.5}$PD was between $O_3$SPD and $PM_{2.5}$SPD.

It is worth noting that the diurnal variations of $SO_4^{2-}$ were different from those of

other aerosol species, with the highest values at 20:00, 9:00, and 16:00 LT in $O_3$SPD,
$PM_{2.5}$SPD, and $O_3$&$PM_{2.5}$PD, respectively, and the lowest values in early morning and
night (5:00 LT in $O_3$SPD and $O_3$&$PM_{2.5}$PD, 23:00 LT in $PM_{2.5}$SPD). For the diurnal
variation of $O_3$, the highest values occurred during the daytime (16:00 LT in $O_3$SPD
and $O_3$&$PM_{2.5}$PD, 15:00 LT in $PM_{2.5}$SPD) and the lowest values appeared at 5:00 LT
in all the cases. Therefore, in $O_3$&$PM_{2.5}$PD, the time of the highest value of $SO_4^{2-}$ was
the same as that of $O_3$, indicating that $SO_4^{2-}$ and $O_3$ were produced synergistically
during the daytime with strong atmospheric oxidation.

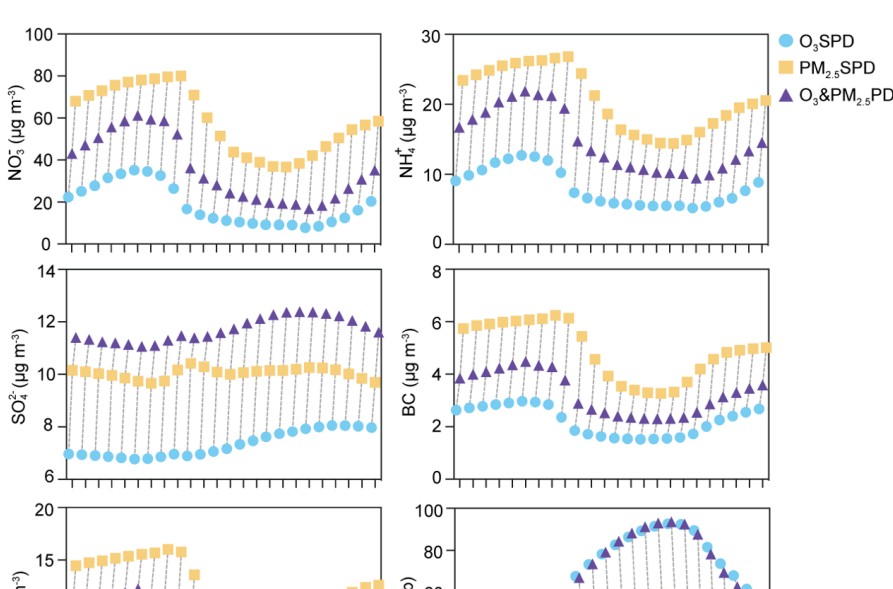


**Figure 7.** The hourly concentrations of $NO_3^-$, $NH_4^+$, $SO_4^{2-}$, BC, OC, and $O_3$ averaged over the model-captured $O_3$SPD, $PM_{2.5}$SPD, and $O_3$ &$PM_{2.5}$PD in BTH in the months of April to October of 2013-2020.

### 3.3.3 Vertical distributions of $O_3$ and $PM_{2.5}$ in $O_3$SPD, $PM_{2.5}$SPD, and $O_3$&$PM_{2.5}$PD

The simulated vertical distributions of $O_3$ and $PM_{2.5}$ averaged over the 18 grids of BTH and the $O_3$SPD, $PM_{2.5}$SPD, and $O_3$&$PM_{2.5}$PD in warm seasons of 2013-2020 are shown in Fig. 8. The vertical distribution of $O_3$ in $O_3$SPD was similar to that in $O_3$&$PM_{2.5}$PD (Fig. 8a). In these two cases, concentrations of $O_3$ increased from the surface to about 975 hPa, remained high between 975 and 819 hPa, and decreased with altitude between 819 and 663 hPa. Although the magnitudes of $O_3$ were close at the surface (61.9 ppbv in $O_3$&$PM_{2.5}$PD and 58.1 ppbv in $O_3$SPD), the concentration of $O_3$



averaged over 975 and 819 hPa was 10.4% higher in $O_3$&$PM_{2.5}$PD than in $O_3$SPD,
which was a very unique feature of $O_3$&$PM_{2.5}$PD. For the case of $PM_{2.5}$SPD, the
concentrations of $O_3$ were the lowest among the three cases and increased gently with
altitude above 975 hPa.

Figure 8b shows the vertical distributions of $PM_{2.5}$ components. In all the cases,

$PM_{2.5}$ concentrations were the highest at the surface, and decreased with altitude from
the surface to 975 hPa. However, concentrations of $PM_{2.5}$ were quite stable between
975 and 819 hPa for $O_3$SPD (about 36.4 $\mu g\ m^{-3}$) and $O_3$&$PM_{2.5}$PD (about 58.1 $\mu g\ m^{-3}$),
corresponding to the stable $O_3$ levels at these altitudes in these two cases (Fig. 8a). For
$PM_{2.5}$SPD, while $PM_{2.5}$ concentration at the surface was the highest among the three
cases, it decreased rapidly between 975 and 819 hPa. The averaged $PM_{2.5}$ concentration
between 975 and 819 hPa was 52.4 $\mu g\ m^{-3}$ in $PM_{2.5}$SPD, which was lower than that in
$O_3$&$PM_{2.5}$PD.

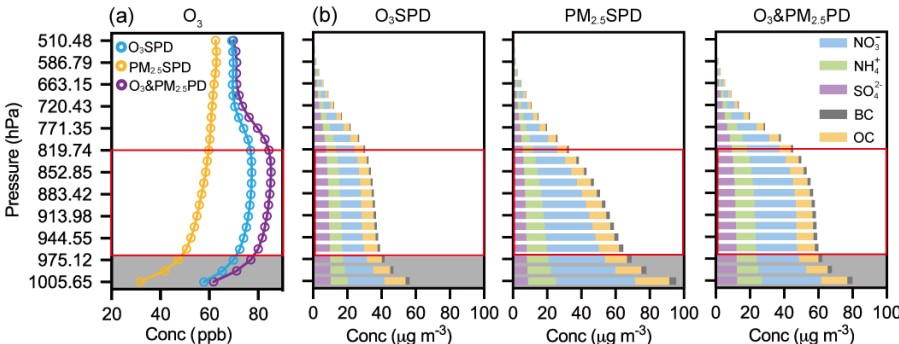


**Figure 8.** The vertical distributions of (a) concentrations of $O_3$ (ppb) and (b) $PM_{2.5}$
components ($\mu g\ m^{-3}$) of $NO_3^-$, $NH_4^+$, $SO_4^{2-}$, BC, OC averaged over the model-
captured $O_3$SPD, $PM_{2.5}$SPD, and $O_3$&$PM_{2.5}$PD in BTH in the months of April to
October of 2013-2020.

To further investigate the differences in vertical profiles of $NO_3^-$, $NH_4^+$, $SO_4^{2-}$, BC,





OC, and PM$_{2.5}$ in O$_3$SPD, PM$_{2.5}$SPD, and O$_3$&PM$_{2.5}$PD, the ratios of concentration at
975 hPa to that at the surface as well as the concentration at 819 hPa to that at 975 hPa
are shown in Table 1. The concentration of PM$_{2.5}$ decreased largely, with the ratio of
PM$_{2.5(975\,hPa)}$/ PM$_{2.5(1005\,hPa)}$ of 0.78 in O$_3$&PM$_{2.5}$PD and of 0.74 in PM$_{2.5}$SPD. For each
of the PM$_{2.5}$ components, the ratios near the surface (from 1005 to 975 hPa, gray shaded
area in Fig. 8) were close in the three types of pollution. While the ratios of NO$_3^-$, NH$_4^+$,
BC, OC were in the range of 0.65-0.80, the ratios of SO$_4^{2-}$ were about 0.93-0.98,
indicating that SO$_4^{2-}$ concentrations were quite uniform from the surface to 975 hPa in
all three types of pollution.
**Table 1.** The ratios at 975 and 1005 hPa (gray shaded area in Fig. 8) and at 819 and
975 hPa (red frame in Fig. 8) of NO$_3^-$, NH$_4^+$, SO$_4^{2-}$, BC, OC, and PM$_{2.5}$ in O$_3$SPD,
PM$_{2.5}$SPD, and O$_3$&PM$_{2.5}$PD in BTH region.

| | | NO$_3^-$ | NH$_4^+$ | SO$_4^{2-}$ | BC | OC | PM$_{2.5}$ |
|---|---|---|---|---|---|---|---|
| Conc$_{819\,hPa}$/ Conc$_{975\,hPa}$ | O$_3$SPD | 0.95 | 0.90 | 0.85 | 0.73 | 0.73 | 0.86 |
| | PM$_{2.5}$SPD | 0.64 | 0.68 | 0.81 | 0.64 | 0.63 | 0.67 |
| | O$_3$&PM$_{2.5}$PD | 0.94 | 0.91 | 0.87 | 0.79 | 0.77 | 0.89 |
| Conc$_{975hPa}$/ Conc$_{1005\,hPa}$ | O$_3$SPD | 0.65 | 0.77 | 0.96 | 0.69 | 0.70 | 0.74 |
| | PM$_{2.5}$SPD | 0.72 | 0.76 | 0.93 | 0.67 | 0.65 | 0.73 |
| | O$_3$&PM$_{2.5}$PD | 0.72 | 0.80 | 0.98 | 0.76 | 0.73 | 0.78 |


In the upper layers (975-819 hPa, red rectangle in Fig. 8), the changes in
concentrations of pollutants with altitude in PM$_{2.5}$SPD were quite different from those
in O$_3$&PM$_{2.5}$PD and O$_3$SPD. The decline of PM$_{2.5}$ from 975 to 819 hPa was slow in
O$_3$&PM$_{2.5}$PD (PM$_{2.5(819\,hPa)}$/PM$_{2.5(975\,hPa)}$=0.89) and O$_3$SPD (0.86) and fast in PM$_{2.5}$SPD



(0.67). Considering that the variation of BC with altitude was mainly driven by
meteorology (Sun et al., 2020), the vertical variations of other components that differed
significantly from BC indicated the influences of chemical processes. In $PM_{2.5}SPD$,
$NO_3^-$, $NH_4^+$, OC had about the same ratio as BC (0.64) (with large decreases with
height), except for $SO_4^{2-}$ concentration that had a ratio of 0.81. In $O_3\&PM_{2.5}PD$, the
ratios of $NO_3^-$, $NH_4^+$, $SO_4^{2-}$ were, 0.94, 0.91, 0.87, respectively, which were much
higher than the value of BC (0.79), indicating $NO_3^-$, $NH_4^+$, $SO_4^{2-}$ were quite uniform
in the layers of 975-819 hPa with the influence of chemical processes, which will be
discussed further in Sect. 3.3.4 below.
**3.3.4 Process analyses for $O_3SPD$, $PM_{2.5}SPD$, and $O_3\&PM_{2.5}PD$**
The process analysis (PA) is applied to identify the relative importance of
atmospheric processes in the three types of pollution. Figure 9 shows the net changes
in $O_3$, $NO_3^-$, $NH_4^+$, $SO_4^{2-}$ by the processes of chemical production (Chem), horizontal
advection (Horizontal_adv), vertical advection (Vertical_avd), and diffusion (Diff,
vertical PBL mixing process) in the GEOS-Chem model, as well as the total of all these
processes (i.e., Chem + Diff + Horizontal_avd + Vertical_avd) in $O_3SPD$, $PM_{2.5}SPD$,
and $O_3\&PM_{2.5}PD$.
For $O_3$, the net changes of $O_3$ by all processes were positive at altitudes of 975-
819 hPa in $O_3\&PM_{2.5}PD$ and $O_3SPD$, in which Chem had the largest positive
contribution (about 1.5 Gg $d^{-1}$), indicating $O_3$ is chemically produced at these layers.
For $NO_3^-$ and $NH_4^+$, the nets of all processes increased mass concentrations at 913-819
hPa in $O_3\&PM_{2.5}PD$ and $O_3SPD$, in which Chem and Vertical_avd were positive and
Chem had the largest positive contribution. The vertical profiles of Chem were similar
for $NO_3^-$ and $NH_4^+$, both of which had the largest positive values at 913-819 hPa ( 2.83
Gg $d^{-1}$ for $NO_3^-$ and 0.88 Gg $d^{-1}$ for $NH_4^+$), leading to higher concentrations of $NO_3^-$





and $NH_4^+$ in $O_3$&$PM_{2.5}$PD than in $O_3$SPD and $PM_{2.5}$SPD. For $SO_4^{2-}$, Chem was
positive from the surface to 510 hPa with a peak around 819 hPa, resulting in the
uniform $SO_4^{2-}$ concentrations at these altitudes as shown in Fig. 8. Chem for $SO_4^{2-}$ was
the highest around 819 hPa in $O_3$&$PM_{2.5}$PD, which was related to the strong liquid-
phase chemical formation of $SO_4^{2-}$ (Fig. S6). In addition to Chem, Vertical_avd also
had positive contributions to the net changes in $O_3$, $NO_3^-$, $NH_4^+$, and $SO_4^{2-}$ at 944-819
hPa. Vertical_avd was negative at 819 hPa and positive between 944 to 819 hPa,
implying that the pollutants were transported from 819 hPa to 944 hPa in $O_3$&$PM_{2.5}$PD.

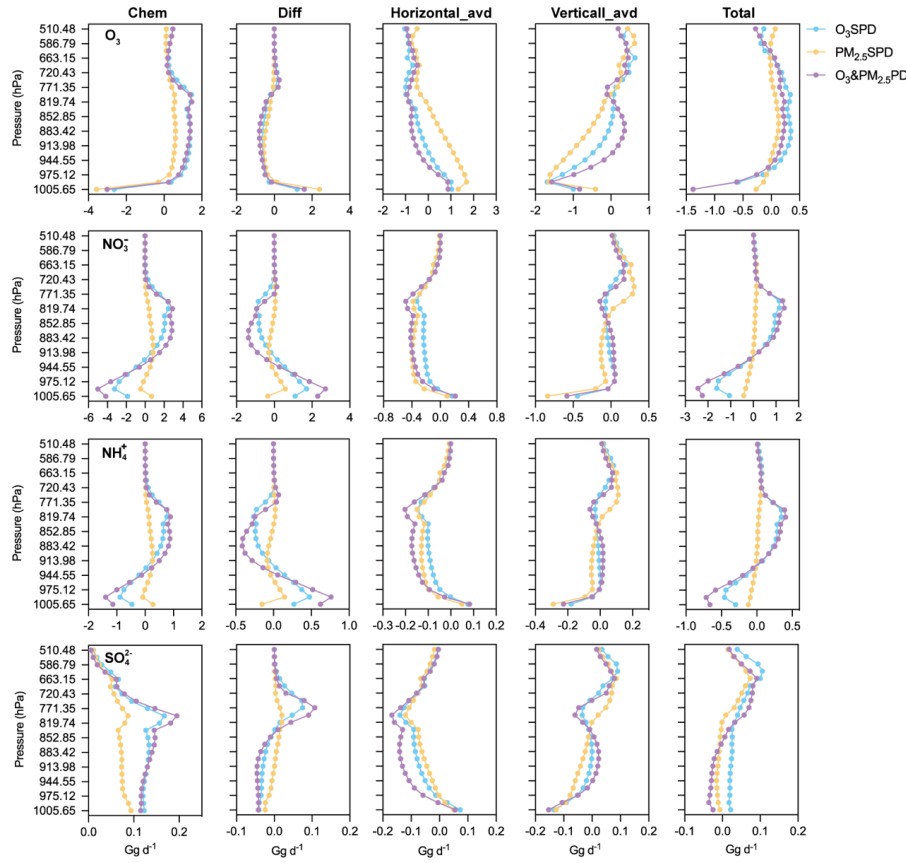


**Figure 9.** The vertical profiles of net changes in $O_3$, $NO_3^-$, $NH_4^+$, and $SO_4^{2-}$ (Gg d$^{-1}$)
over BTH by each and total of processes. The values were averaged over the model-





captured regional $O_3SPD$, $PM_{2.5}SPD$, and $O_3\&PM_{2.5}PD$ in April-October of 2013-

2020.

Overall, $NO_3^-$, $NH_4^+$, and $SO_4^{2-}$ all had larger chemical productions at 913-819

hPa in $O_3\&PM_{2.5}PD$ compared to those in $O_3SPD$ and $PM_{2.5}SPD$, accompanied by
strong vertical transport from 819 hPa to near the surface, resulting in the quite uniform
vertical profiles at 975-819 hPa in $O_3\&PM_{2.5}PD$. In addition, the vertical profiles of net
changes in $PM_{2.5}$ over BTH are shown in Fig. S7 for these three cases. Since $NO_3^-$,
$NH_4^+$, and $SO_4^{2-}$ were the major components of $PM_{2.5}$, the PA of $PM_{2.5}$ is similar to that
of each component.
**3.3.5 Observed AOD in $O_3SPD$, $PM_{2.5}SPD$, and $O_3\&PM_{2.5}PD$**

To try to support the model result that $O_3\&PM_{2.5}PD$ had more uniform vertical

profile than $PM_{2.5}SPD$ from the surface to 819 hPa altitude, we present the scatter plots
of observed AOD (at 440 nm and 675 nm) versus observed $PM_{2.5}$ concentrations in
$O_3SPD$, $PM_{2.5}SPD$, and $O_3\&PM_{2.5}PD$ in Fig. 10. AERONET observations of AOD
from 2013 to 2020 are available at three sites in BTH (that is, Beijing (39.97°N, 116.38°
E), Beijing-CAMS (39.93°N, 116.31°E), Xianghe (39.75°N, 116.96°E)). At Beijing
(39.97°N, 116.38°E), AOD (440nm and 675nm) increased with $PM_{2.5}$ concentration in
all three types of pollution. However, under the same levels of surface $PM_{2.5}$
concentration, AOD values in $O_3\&PM_{2.5}PD$ were higher than in $PM_{2.5}SPD$, implying
that the column burdens of aerosols were generally higher in $O_3\&PM_{2.5}PD$ than in
$PM_{2.5}SPD$, which may support the unique vertical distribution of $PM_{2.5}$ in $O_3\&PM_{2.5}PD$
shown in Fig. 8b. The scatter plots at Beijing-CAMS and Xianghe sites are similar and
are shown in Fig. S8.





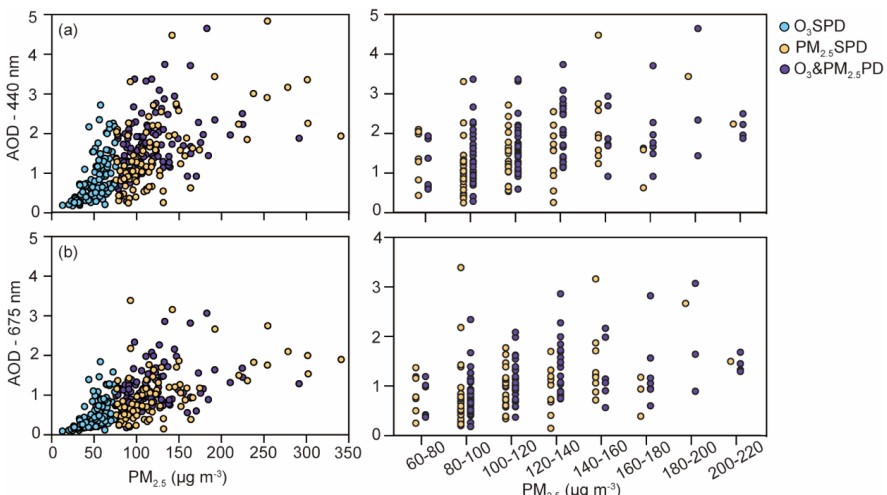

**Figure 10.** The scatterplots of (a) AOD (440 nm) and (b) AOD (675 nm) versus

observed $PM_{2.5}$ concentrations in $O_3SPD$, $PM_{2.5}SPD$, and $O_3\&PM_{2.5}PD$ in Beijing

(39.97°N, 116.38°E) in April-October of 2013-2020.

**3.4 Meteorological conditions for $O_3SPD$, $PM_{2.5}SPD$, and $O_3\&PM_{2.5}PD$ over BTH**

Figure 11 shows the vertical profiles of RH, T, and OMEGA for $O_3SPD$, $PM_{2.5}SPD$,

and $O_3\&PM_{2.5}PD$ captured by the model over BTH in the months of April to October

form 2013-2020. It should be noted that $O_3\&PM_{2.5}PD$ had an unique vertical

distribution of RH. Near the surface, the values of RH in $O_3\&PM_{2.5}PD$ were between

those in $O_3SPD$ and $PM_{2.5}SPD$. However, in the upper layers (883-771 hPa),

$O_3\&PM_{2.5}PD$ had the highest RH among the three cases with a peak value of 58.2%.

As a result, the strongest aqueous chemical production of $SO_4^{2-}$ (aqueous oxidation of

$SO_2$ by $H_2O_2$) occurred in $O_3\&PM_{2.5}PD$ around 819 to 771 hPa (Fig. S6). The vertical

profiles of temperature were similar in the three types of pollution, with the lowest

temperature in $PM_{2.5}SPD$. The vertical profiles of OMEGA were different in the three

cases. In $O_3SPD$ and $O_3\&PM_{2.5}PD$, OMEGA had positive values around 819 hPa,

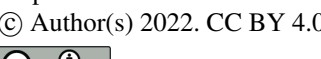



indicating a strong sinking airflow, leading to a downward transport of pollutants.
Under $O_3$&$PM_{2.5}$PD, the average values of PBLH and SWGDN were 946.1 m and
257.2 W m$^{-2}$, respectively, which were higher (lower) than those in $PM_{2.5}$SPD ($O_3$SPD)
(Fig. S9).

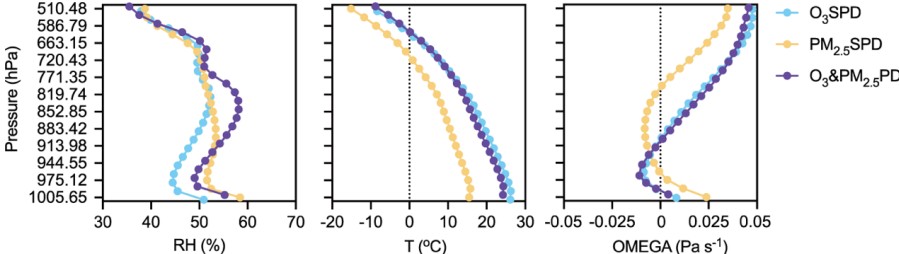


**Figure 11.** The vertical profiles of RH (%), T (°C), and OMEGA (Pa s$^{-1}$) averaged over
BTH and over the model-captured regional $O_3$SPD, $PM_{2.5}$SPD, and $O_3$&$PM_{2.5}$PD in in
April-October of 2013-2020.

Figure 12 shows the composited weather patterns for regional $O_3$SPD, $PM_{2.5}$SPD,

and $O_3$&$PM_{2.5}$PD (over 50% cities in the BTH experiencd the pollution) that were
captured by the model in April-October of 2013-2020. The weather patterns of
$O_3$&$PM_{2.5}$PD were similar to some extent to those of $O_3$SPD but were quite different
from those of $PM_{2.5}$SPD. In $O_3$&$PM_{2.5}$PD, the BTH region was controlled by westerlies
at 500 hPa without cold air intrusion from the north, and was under the high pressure
ridge of the Western Pacific Subtropical High (WPSH) at 850 hPa. The strong
southerlies at 850 hPa brought moist air from the south (Fig. S10), resulting in a high
RH that was beneficial to the aqueous chemical production of $SO_4^{2-}$ in $O_3$&$PM_{2.5}$PD.
In $O_3$SPD, BTH was also under the influence of the high pressure ridge of the WPSH
at 850 hPa, but it was weaker than in $O_3$&$PM_{2.5}$PD. Besides, the Northeast Cold Vortex
(NCV) was located to the southwest of BTH at 850 hPa in $O_3$SPD, leading to dry and
warm conditions, which was favorable for the formation of $O_3$. In $PM_{2.5}$SPD, the BTH
region was under the influence of both the continental high and the WPSH at 850 hPa.



At the surface, BTH was under the influence of a uniform high pressure with very weak
winds and hence stagnate atmosphere, which was conducive to the accumulation of
$PM_{2.5}$.

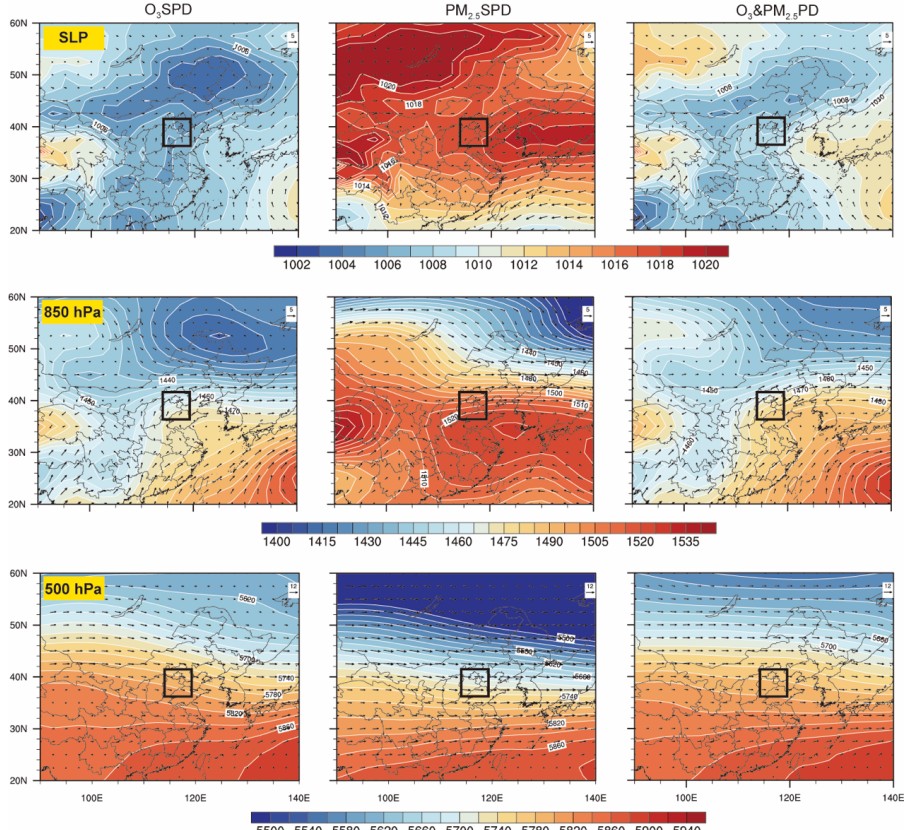


**Figure 12.** Composites of wind field (m s⁻¹) with SLP (sea level pressure) and with
geopotential height at 850 hPa and 500 hPa for regional $O_3$SPD, $PM_{2.5}$SPD, and
$O_3$&$PM_{2.5}$PD that were captured by the model in April-October of 2013-2020. The
solid black rectangle indicates BTH region.

**4. Conclusions**

We used the observed hourly concentrations of $O_3$ and $PM_{2.5}$ from CNEMC and

the model results from the nested-grid version of the GEOS-Chem model to examine



the chemical and physical characteristics of the co-polluted days by $O_3$ and $PM_{2.5}$
($O_3$&$PM_{2.5}$PD) over the BTH region for eight warm seasons (April-October) from
2013 to 2020. The characteristic of $O_3$&$PM_{2.5}$PD were compared with those of the
polluted days by $O_3$ alone ($O_3$SPD) and by $PM_{2.5}$ alone ($PM_{2.5}$SPD). In April-October
of 2013-2020, the observed $O_3$SPD, $PM_{2.5}$SPD, and $O_3$&$PM_{2.5}$PD were 2954, 2148,
and 1614 days, respectively, in which 75.0% (2954/3937), 58.1% (2148/3698), and
79.7% (1614/2024) were captured by the GEOS-Chem model, respectively. We
carried out composited analyses of the chemical and physical characteristics for
$O_3$SPD, $PM_{2.5}$SPD, and $O_3$&$PM_{2.5}$PD by using the samples (days) captured by both
the observations and the model.

The chemical characteristics of $O_3$&$PM_{2.5}$PD were found to be different from

those of $O_3$SPD, $PM_{2.5}$SPD at the surface. $O_3$&$PM_{2.5}$PD occurred with high levels of
atmospheric oxidants (high OH and $O_x$), with higher SOR and NOR compared to those
in $O_3$SPD and $PM_{2.5}$SPD, leading to high concentrations of both $O_3$ and $PM_{2.5}$. At the
surface, the composited concentrations of $NO_3^-$, $NH_4^+$, BC, and OC were the highest in
$PM_{2.5}$SPD, while the composited concentration of $SO_4^{2-}$ was the highest in
$O_3$&$PM_{2.5}$PD. There was a strong formation of $SO_4^{2-}$ during the daytime in
$O_3$&$PM_{2.5}$PD in the oxidative atmosphere.

We also found unique features of the vertical distributions of $O_3$ and $PM_{2.5}$ in

$O_3$&$PM_{2.5}$PD. Concentrations of $PM_{2.5}$ were stable and high between 975 and 819 hPa
in $O_3$&$PM_{2.5}$PD, unlike those in $PM_{2.5}$SPD that decreased rapidly with the altitude. In
$O_3$&$PM_{2.5}$PD, the vertical profiles of $NO_3^-$, $NH_4^+$, and $SO_4^{2-}$ were quite uniform at
975-819 hPa, corresponding to the stable $O_3$ concentrations at these altitudes. The
process analysis (PA) showed that $NO_3^-$, $NH_4^+$, and $SO_4^{2-}$ all had larger chemical
productions at altitudes of 913-819 hPa in $O_3$&$PM_{2.5}$PD compared to those in $O_3$SPD




and PM$_{2.5}$SPD. The chemical production of SO$_4^{2-}$ had large positive values from the
surface to about 500 hPa. The Vertical_avd also had positive contributions to the net
changes in O$_3$, NO$_3^-$, NH$_4^+$, and SO$_4^{2-}$ at 944-819 hPa in O$_3$&PM$_{2.5}$PD. Therefore, the
strong chemical productions at 913-819 hPa accompanied by the downward transport
resulted in the quite uniform vertical profiles at 975-819 hPa in O$_3$&PM$_{2.5}$PD.
Figure 13 summarizes the chemical and physical characteristics in O$_3$&PM$_{2.5}$PD,
O$_3$SPD, and PM$_{2.5}$SPD in the BTH region. In O$_3$&PM$_{2.5}$PD, the strong chemical
productions of O$_3$, NO$_3^-$, NH$_4^+$, and SO$_4^{2-}$ occur at high altitudes of 913-819 hPa where
RH is high, and the accompanied downward airflow causes the stable concentrations at
944-819 hPa. The composited PBLH in O$_3$&PM$_{2.5}$PD is about 946.1 m, and the strong
mixed diffusion underneath the PBLH leads to high concentrations of pollutants at the
ground level. In contrast, O$_3$SPD occurs in hot and dry atmosphere with composited
PBLH of 1073.5 m. Strong O$_3$ chemical production occurs around 819 hPa, and O$_3$ is
then transported to the surface by downward air flow. The atmosphere is stable and
stagnate when PM$_{2.5}$SPD occurs, with the lowest PBLH of 681.8 m. High RH (high
chemical formation of PM$_{2.5}$) and the accumulation of aerosols lead to the highest
surface-layer PM$_{2.5}$ in PM$_{2.5}$SPD.

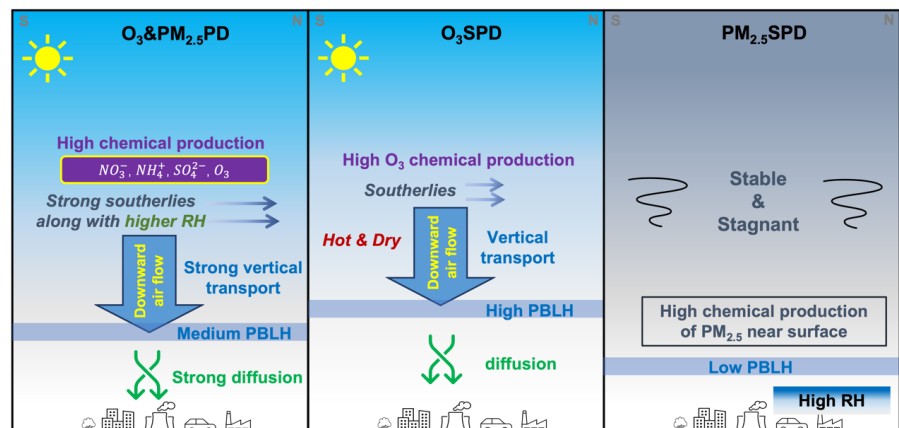





**Figure 13.** A schematic diagram of chemical and physical and characteristics in
$O_3SPD$, $PM_{2.5}SPD$, and $O_3\&PM_{2.5}PD$ in BTH region.

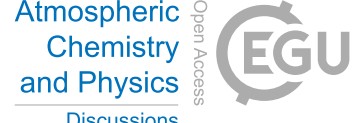

***Data availability.***
The GEOS-Chem model is available at https://geos-chem.seas.harvard.edu (last
access: 5 August 2022). The observed hourly surface concentrations of air pollutants
are derived from the China National Environ- mental Monitoring Center
(https://air.cnemc.cn:18007/, CNEMC, 2022). The simulation results are available
upon request from the corresponding author (hongliao@nuist.edu.cn).

***Author contributions.***
HD and HL conceived the study and designed the experiments. HD performed model
simulations and analysed the data. KL, XY, YY, JZ, JJ, and BL provided useful
comments on the paper. HD and HL prepared the paper, with contributions from all co-
authors.

***Competing interests.***
The authors declare that they have no conflict of interest.

***Acknowledgements.***
We acknowledge the CNEMC for making their data publicly available. We
acknowledge the efforts of GEOS-Chem working groups for developing and
managing the model.

***Financial support.***



This work was supported by the National Natural Science Foundation of China (Grant
No. 42021004) and the National Key Research and Development Program of China
(Grant No. 2019YFA0606800)



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
