# Peer review of "Composited analyses of the chemical and physical characteristics of co- polluted days by ozone and PM$_{2.5}$ over 2013-2020 in the Beijing–Tianjin–Hebei region"

_Atmospheric Chemistry and Physics, 2022_

## Author Comment (AC1)

**Response to Comments of Reviewer #1**

**Manuscript number:** acp-2022-557

**Title:** Composited analyses of the chemical and physical characteristics of co-polluted days by ozone and PM$_{2.5}$ over 2013–2020 in the Beijing–Tianjin–Hebei region

**General comments**:

In this manuscript the chemical and physical characteristics of O$_3$ (O$_3$SPD), PM$_{2.5}$ (PM$_{2.5}$SPD) polluted days and O$_3$ and PM$_{2.5}$ (O$_3$&PM$_{2.5}$PD) co-polluted days over BTH are investigated by using the 3-D global chemical transport model (GEOS-Chem). This manuscript is clearly written and well organized, and its conclusions are interesting.

Thanks to the referee for the helpful comments and constructive suggestions. We have revised the manuscript carefully and the point-to-point responses are listed below.

**Major concerns/questions:**

1.  The simulated PM$_{2.5}$ components including NO$_3^-$, NH$_4^+$, SO$_4^{2-}$, BC, and OC are compared against observed PM$_{2.5}$ concentrations, and the comparison shows that the simulated PM$_{2.5}$ had a NMB of -26.9%. Even with the adjusted thresholds, percentages of observed polluted days for PM$_{2.5}$SPD shown in Figure c are lower than for O$_3$SPD and O$_3$&PM$_{2.5}$PD. Is the underestimation attributable to some missing primary aerosols?

**Response:**

Thanks for pointing this out. In this work we calculated PM$_{2.5}$ concentration as the sum of the simulated masses of SO$_4^{2-}$, NO$_3^-$, NH$_4^+$, BC, and OC as in previous studies (*Yang et al., 2016; Dang et al., 2019; Xie et al., 2022*). Although primary aerosols such as mineral dust and sea salt are not the dominant aerosol species in China (*Xuan et al., 2000; Ye et al., 2003; Duan et al., 2006; Zhao et al., 2013*), the absence of them in the calculation of PM$_{2.5}$ contributed to the low biases in simulated PM$_{2.5}$SPD. To clarify this, we have added the following sentences in the first paragraph of Section 3.2.1: "It should be noted that mineral dust and sea-salt aerosols were not considered in this study, because they are not the major aerosol components in China and the concentrations are generally low based on previous measurements (*Xuan et al., 2000; Ye et al., 2003; Duan et al., 2006; Zhao et al., 2013*). However, excluding dust and sea salt may lead to low biases in simulated PM$_{2.5}$ concentrations.".

2.  In the analysis two oxidation indicators (sulfur oxidation ratio and nitrogen oxidation ratio) are used, but not assessed. As observed SO$_2$ and NO$_2$ concentrations are available at CNEMC, model performance for SO$_2$ and NO$_2$ is suggested to be evaluated.

**Response:**

Following the Reviewer's suggestion, we have added Figure S2 in the Supplementary Material to evaluate the model performance for SO$_2$ and NO$_2$. We have also added the following sentences in the second paragraph of Section 3.2.1 to describe the model

performance: "Due to the lack of the publicly accessible long-term observations of $PM_{2.5}$ components in China, we compared the simulated $SO_2$ and $NO_2$ (precursors for $SO_4^{2-}$ and $NO_3^-$) with observations from CNEMC in Fig. S2. The simulated daily mean concentrations of $NO_2$ ($SO_2$) agree well with the observations from CNEMC with R of 0.82 (0.78) and MB of -14.9% (9.3%)."

[Figure]

**Figure S2.** Simulated and observed daily mean surface-layer concentrations of $NO_2$ and $SO_2$ as well as the scatterplots of simulated versus observed values for April to October of 2013–2020 in BTH.

3.  Figure S5a shows the hourly variations of PBLH (m) averaged in all model-captured $O_3$SPD (blue), $PM_{2.5}$SPD (yellow), and $O_3$&$PM_{2.5}$PD (purple). Average PBLH at noon time for $O_3$SPD and $O_3$&$PM_{2.5}$PD is over 2000m, why are they so high? Figure S5b shows the daily anomaly of PBLH for $O_3$SPD and $O_3$&$PM_{2.5}$PD at night time exceeds -500m, while at noon time over 1000m. How does PBLH usually change over BTH?

**Response:**
The hourly variations of PBLH were taken from MERRA-2. Hourly PBLH values averaged over all days in the warm months (April-October) of 2013-2020 in BTH are shown in Figure R1. PBLH reached the highest value of 1842 m at 14:00 and the lowest value of 300 m at 6:00. The hourly variation of PBLH over BTH from MERRA-2 agrees closely with observations. *Guo et al. (2016)* used the fine-resolution sounding observations of 391552 profiles from January 2011 to July 2015 across China and reported that PBLH in BTH averaged over summers was around 1600-2000 m at 14:00 and 100-350 m at 2:00. *Tang et al. (2021)* carried out two field campaigns in Shijiazhuang, the area with serious air pollution in BTH, from August 25 to September 19, 2018 (late summer) and from June 8 to July 2, 2019 (early summer). They found that the highest PBLH during the daytime in summer was around 1800 m in $O_3$ polluted days.

[Figure]

**Figure R1.** Hourly variation of PBLH from MERRA2 averaged over all days in the warm months (April-October) of 2013-2020 in BTH.

4. Figure S6 shows the vertical profile of $SO_4^{2-}$ chemical production. Why is $SO_4^{2-}$ chemical production larger at high levels than at low levels? Is it associated with cloud or high relative humidity? How is $SO_2$ concentration distributed vertically? How to understand the difference between $O_3SPD$ and $PM_{2.5}SPD$?

**Response:**
The vertical profile of $SO_4^{2-}$ chemical production from aqueous oxidation shown in Figure S6 was associated with cloud and relative humidity (RH) (see Figures R2(a)-R2(c) below). The strongest aqueous chemical production of $SO_4^{2-}$ occurred around 819 to 771 hPa where cloud optical depth (OPTDEPTH) and RH peaked. Clouds and high RH were conductive to the aqueous chemical production of $SO_4^{2-}$. As a result, $SO_4^{2-}$ chemical production was larger at high levels than at low levels.

Figure R2(d) below shows the vertical distributions of $SO_2$ in the model-captured $O_3SPD$, $PM_{2.5}SPD$, and $O_3\&PM_{2.5}PD$. For all the cases, $SO_2$ concentrations were the highest at the surface, decreased rapidly with height from the surface to 975 hPa, and then declined slowly at 944-819 hPa.

The differences between $O_3SPD$ and $PM_{2.5}SPD$ in Figure S6 indicated in the differences in vertical distribution of the chemical production of $SO_4^{2-}$, which have been explained in the first paragraph of this response.

[Figure]

**Figure R2.** The vertical distributions of (a) cloud optical thickness (OPTDEPTH), (b) relative humidity (RH), (c) $SO_4^{2-}$ chemical production from aqueous oxidation ($P(SO_4^{2-})$), and (d) $SO_2$ averaged over the model-captured $O_3SPD$, $PM_{2.5}SPD$, and $O_3\&PM_{2.5}PD$ in BTH in the months of April to October of 2013-2020.

5. It is interesting to see Figure 8a that $O_3$ levels for $O_3SPD$ are lower than for $O_3\&PM_{2.5}PD$. Does it mean high $PM_{2.5}$ leads to increase in $O_3$? Figure 8b also shows BC is well mixed vertically up to ~819 hPa. Is it an average for all selected days?

**Response:**

In this study, we found that high $O_3$ levels lead to increases in $PM_{2.5}$ in $O_3\&PM_{2.5}PD$ for the following reasons: (1) $O_3\&PM_{2.5}PD$ occurred with high levels of atmospheric oxidants (OH and $O_x$), high SOR and NOR, leading to high concentrations of $PM_{2.5}$, (2) the vertical profiles of $NO_3^-$, $NH_4^+$, and $SO_4^{2-}$ were quite uniform at 975-819 hPa, corresponding to the stable $O_3$ concentrations at these altitudes, as confirmed by the process analysis.

However, high $PM_{2.5}$ concentrations usually lead to reductions in $O_3$, as reported by previous studies. *Li et al. (2019)* reported that $PM_{2.5}$ influences $O_3$ chemical production through the uptake of both $HO_x$ and $NO_x$ radicals, which suppressed $O_3$ production under high $PM_{2.5}$ ($PM_{2.5} > 60$ $\mu g$ $m^{-3}$) in the summer of 2013-2018 in the North China Plain. *Yang et al. (2022)* found that the aerosol-photolysis and aerosol-radiation interactions caused significant decreases in daytime surface-layer $O_3$ concentrations during multi-pollutant air pollution episodes.

Since the concentrations of BC were small and not easily visible in Figure 8b, we present the vertical distributions of BC in Figure R3 below, which are averaged over all selected samples of the three types of pollution (i.e., 2954 samples of $O_3SPD$, 2148 samples of $PM_{2.5}SPD$, and 1614 samples of $O_3\&PM_{2.5}PD$). The variation of BC with

altitude was mainly driven by meteorology. The ratio of concentration at 819 to that at 975 hPa for BC and other $PM_{2.5}$ components is shown in Table 1 of our manuscript for $O_3$SPD, $PM_{2.5}$SPD, and $O_3\&PM_{2.5}$PD. The $BC_{(819\ hPa)}$ /$BC_{(975\ hPa)}$ was 0.73, 0.64, and 0.79 in $O_3$SPD, $PM_{2.5}$SPD, and $O_3\&PM_{2.5}$PD, respectively. In $PM_{2.5}$SPD, $NO_3^-$, $NH_4^+$, OC had about the same ratio as BC (0.64) (with large decreases with height), except for $SO_4^{2-}$ concentration that had a ratio of 0.81. In $O_3\&PM_{2.5}$PD, the ratios of $NO_3^-$, $NH_4^+$, $SO_4^{2-}$ were, 0.94, 0.91, 0.87, respectively, which were much higher than the value of BC (0.79), indicating $NO_3^-$, $NH_4^+$, $SO_4^{2-}$ were quite uniform in the layers of 975-819 hPa with the influence of chemical processes. We have discussed these characteristics in Section 3.3.4.

[Figure]

**Figure R3.** The vertical distributions of BC averaged over the model-captured $O_3$SPD, $PM_{2.5}$SPD, and $O_3\&PM_{2.5}$PD in BTH in the months of April to October of 2013-2020.

**Table 1.** The ratio of concentration at 819 to that at 975 hPa for $NO_3^-$, $NH_4^+$, $SO_4^{2-}$, BC, OC, and $PM_{2.5}$ averaged over $O_3$SPD, $PM_{2.5}$SPD, and $O_3\&PM_{2.5}$PD in BTH.

| | | $NO_3^-$ | $NH_4^+$ | $SO_4^{2-}$ | BC | OC | $PM_{2.5}$ |
|---|---|---|---|---|---|---|---|
| $Conc_{819\ hPa}$/ $Conc_{975\ hPa}$ | $O_3$SPD | 0.95 | 0.90 | 0.85 | 0.73 | 0.73 | 0.86 |
| | $PM_{2.5}$SPD | 0.64 | 0.68 | 0.81 | 0.64 | 0.63 | 0.67 |
| | $O_3\&PM_{2.5}$PD | 0.94 | 0.91 | 0.87 | 0.79 | 0.77 | 0.89 |

**References:**

Dang, R. J., and Liao, H.: Radiative forcing and health impact of aerosols and ozone in China as the consequence of clean air actions over 2012-2017, Geophys. Res. Lett., 46, 12511-12519, https://doi:10.1029/2019GL084605, 2019.

Duan, F., He, K., Ma, Y., Yang, F., Yu, X., Cadle, S. H., Chan, T., and Mulawa, P. A.: Concentration and chemical characteristics of PM2.5 in Beijing, China: 2001–2002, Sci. Total Environ., 355(1–3), 264–275, https://doi:10.1016/j.scitotenv.2005.03.001, 2006.

Guo, J. P., Miao, Y. C., Zhang, Y., Liu, H., Li, Z. Q., Zhang, W. C., He, J., Lou, M. Y., Yan, Y., Bian, L., and Zhai, P. M.: The climatology of planetary boundary layer

height in China derived from radiosonde and reanalysis data, Atmos. Chem. Phys., 16, 13309-13319, https://doi:10.5194/acp-16-13309-2016, 2016.

Li, K., Jacob, D. J., Liao, H., Zhu, J., Shah, V., Shen, L., Bates, K., Zhang, Q., and Zhai, S.: A two-pollutant strategy for improving ozone and particulate matter air quality in China, Nature Geoscience, 12, 906-910, https://doi:10.1038/s41561-019-0464-x, 2019.

Tang, G., Liu, Y., Huang, X., Wang, Y., Hu, B., Zhang, Y., Song, T., Li, X., Wu, S., Li, Q., Kang, Y., Zhu, Z., Wang, M., Wang, Y., Li, T., Li, X., and Wang ,Y.: Aggravated ozone pollution in the strong free convection boundary layer, Sci. Total Environ., 20, 788, 147740, https://doi: 10.1016/j.scitotenv, 2021.

Xie, P. F., and Liao, H.: The impacts of changes in anthropogenic emissions over China on PM2.5 concentrations in South Korea and Japan during 2013–2017, Front. Environ. Sci., 10:841285, https://doi:10.3389/fenvs.2022.841285, 2022.

Xuan, J., Liu, G., and Du, K.: Dust emission inventory in northern China, Atmos. Environ., 34(26), 4565–4570, https://doi:10.1016/S1352-2310(00) 00203-X, 2000.

Yang, H., Chen, L., Liao, H., Zhu, J., Wang, W., and Li, X.: Impacts of aerosol-photolysis interaction and aerosol-radiation feedback on surface-layer ozone in North China during multi-pollutant air pollution episodes, Atmos. Chem. Phys., 22, 4101-4116, https://doi:10.5194/acp-22-4101-2022, 2022.

Yang, Y., Liao, H., and Lou, S. J.: Increase in winter haze over eastern China in the past decades: Roles of variations in meteorological parameters and anthropogenic emissions, J. Geophys. Res., 121, 13050-13065, https://doi:10.1002/2016JD025136, 2016.

Ye, B., Ji, X., Yang, H., Yao, X., Chan, C. K., Cadle, S. H., Chan, T., and Mulawa, P. A.: Concentration and chemical composition of PM2.5 in Shanghai for a 1-year period, Atmos. Environ., 37(4), 499–510, https://doi:10.1016/S1352-2310(02)00918-4, 2003.

Zhao, X. J., Zhao, P. S., Xu, J., Meng, W., Pu, W. W., Dong, F., He, D., and Shi, Q. F.: Analysis of a winter regional haze event and its formation mechanism in the North China Plain, Atmos. Chem. Phys., 13, 5685–5696, https://doi:10.5194/acp-13-5685-2013, 2013.

---

## Author Comment (AC2)

**Response to Comments of Reviewer #2**

**Manuscript number:** acp-2022-557
**Title:** Composited analyses of the chemical and physical characteristics of co-polluted days by ozone and $PM_{2.5}$ over 2013–2020 in the Beijing–Tianjin–Hebei region

**General comments**:

In recent years, decreases in $PM_{2.5}$ but increases in $O_3$ over eastern China make the co-occurrences of $PM_{2.5}$ and $O_3$ polluted days ($O_3\&PM_{2.5}PD$) an important issue related to human health. In this work, Dai et al. explored the chemical and synoptic characteristics of $O_3\&PM_{2.5}PD$ in Beijing-Tianjin-Hebei (BTH) region within a GEOS-Chem framework. They provided comprehensive analysis and concrete details in the differences among $PM_{2.5}$ alone polluted days ($PM_{2.5}SPD$), $O_3$ alone polluted days ($O_3SPD$) and $O_3\&PM_{2.5}PD$. Results are novel and of scientific significance. I would like to suggest publication after addressing my comments below:

Thanks to the referee for the helpful comments and constructive suggestions. We have revised the manuscript carefully and the point-to-point responses are listed below.

**Major Concerns:**

1. I suggest authors to separate Section 3.3 into two or three parts, where the chemical characteristics, vertical profile and process analysis are described respectively. The current demonstration looks not very logistic and thus makes it hard to follow.

**Response:**

This paper is focused on the characteristics of chemical composition in $O_3SPD$, $PM_{2.5}SPD$, and $O_3\&PM_{2.5}PD$, thus the description of the chemical characteristics is a little bit long, but logical. To make this section easier to follow, we have added the following sentences in the first paragraph of Section 3.3 to act as a guide to the readers: "In this section, to investigate the chemical characteristics of $O_3SPD$, $PM_{2.5}SPD$ and $O_3\&PM_{2.5}PD$, we present first the simulated atmospheric oxidants in 3.3.1, and then show the simulated surface concentrations and vertical profiles of $PM_{2.5}$ and MDA8 $O_3$ in 3.3.2 and 3.3.3, respectively, followed by the process analysis in 3.3.4. The observed AOD values to verify the model results are presented in 3.3.5.".

2. Section 3.2, GEOS-Chem still significantly underestimates peak $PM_{2.5}$ concentrations as shown in Fig. 3d. Which $PM_{2.5}$ components dominate such underestimates? I'm worried that GEOS-Chem incapacity in simulating peak $PM_{2.5}$ could significantly influence the following analysis related to the differences in $SO_4^{2-}$ and $NO_3^-$ among $O_3SPD$, $PM_{2.5}SPD$ and $O_3\&PM_{2.5}PD$. At least more evaluation and discussions are necessary.

**Response:**

The Reviewer #1 pointed out that the underestimation of $PM_{2.5}$ might be due to the absence of mineral dust and sea salt aerosols in simulated $PM_{2.5}$, therefore we have added the following sentences in the first paragraph of Section 3.2.1: "It should be noted

that mineral dust and sea-salt aerosols were not considered in this study, because they are not the major aerosol components in China and the concentrations are generally low based on previous measurements (*Xuan et al., 2000; Ye et al., 2003; Duan et al., 2006; Zhao et al., 2013*). However, excluding dust and sea salt may lead to low biases in simulated $PM_{2.5}$ concentrations.".

Due to the lack of the publicly accessible long-term observations of $PM_{2.5}$ components in China, we compared the simulated $SO_2$ and $NO_2$ (precursors for $SO_4^{2-}$ and $NO_3^-$) with observations from CNEMC. The evaluation of model performance for $SO_2$ and $NO_2$ in BTH over the eight warm seasons (April to October, 2013–2020) has been added as Figure S2 in the Supplementary Material. We have also added the following sentences in the second paragraph of Section 3.2.1 to describe the model performance: "Due to the lack of the publicly accessible long-term observations of $PM_{2.5}$ components in China, we compared the simulated $SO_2$ and $NO_2$ (precursors for $SO_4^{2-}$ and $NO_3^-$) with observations from CNEMC in Fig. S2. The simulated daily mean concentrations of $NO_2$ ($SO_2$) agree well with the observations from CNEMC with R of 0.82 (0.78) and MB of -14.9% (9.3%).".

[Figure]

**Figure S2.** Simulated and observed daily mean surface-layer concentrations of $NO_2$ and $SO_2$ as well as the scatterplots of simulated versus observed values for April to October of 2013–2020 in BTH.

3. Lines 351-354 and Fig. 6, compared to $O_3$&$PM_{2.5}$PD, less S was oxidized into $SO_4^{2-}$ during $PM_{2.5}$SPD and less N was oxidized into $NO_3^-$ during $O_3$SPD. Such differences also reflected in the $PM_{2.5}$ components in Fig. 6. Are there any explanations about that? In addition, I'm curious what are the dominant oxidation pathways (e.g. $SO_2$ oxidation through $H_2O_2$, $O_3$, OH or $NO_2$) of $SO_2$ and $NO_x$ in GEOS-Chem? Can pathways be different among $O_3$SPD, $PM_{2.5}$SPD and $O_3$&$PM_{2.5}$PD?

**Response:**

In $O_3$SPD, $PM_{2.5}$SPD, and $O_3$&$PM_{2.5}$PD, the averaged values of SOR were 50.0%, 36.7%, and 49.7%, and those of NOR were 55.4%, 70.0%, and 70.2%, respectively. Compared to $O_3$&$PM_{2.5}$PD, SOR was lower in $PM_{2.5}$SPD, which was due to the lower atmospheric oxidation capacity and hence less $SO_4^{2-}$ generation in $PM_{2.5}$SPD. Compared to $O_3$&$PM_{2.5}$PD, NOR was lower in $O_3$SPD because $O_x$ ($O_x = O_3 + NO_2$) was lower in $O_3$SPD. We have clarified these reasons in the second paragraph of Section 3.3.1.

The formation pathways of $SO_4^{2-}$ mainly include the gas-phase oxidation of $SO_2$ by OH and the in-cloud oxidation of $SO_2$ by $O_3$ and $H_2O_2$. The process that influences $NO_3^-$ concentration is the gas-to-aerosol conversion of $HNO_3$ to form $NO_3^-$ in GEOS-Chem as reported by *Mu et al. (2014)*. As for the dominant pathway for $SO_4^{2-}$, the in-cloud $SO_4^{2-}$ formation is known as the major source of global $SO_4^{2-}$ because aqueous-phase oxidation of S(IV) by $H_2O_2$ and $O_3$ occurs much more rapidly than gas-phase oxidation of $SO_2$ by OH (*Barth et al., 2000; Ervens, 2015*). Previous modeling studies concluded that a dominant fraction of $SO_4^{2-}$ (60–90%) is formed via in-cloud aqueous chemistry globally (*Dovrou et al., 2019; Ervens, 2018; Liao et al., 2013; Harris et al., 2013; Ma and Salzen, 2006*).

The GEOS-Chem's fullchem simulation outputs only the in-cloud $SO_4^{2-}$ formation pathway, which is the dominant pathway for $SO_4^{2-}$ as mentioned in the above paragraph. We don't think the pathways were different among $O_3$SPD, $PM_{2.5}$SPD and $O_3$&$PM_{2.5}$PD, but the values of process analyses for chemistry were different because of the different chemical and physical conditions.

4. In Fig.9, I'm confused about the totally different diffusion profile in $SO_4^{2-}$ relative to $NO_3^-$ and $NH_4^+$. In the PBL, air pollutants are supposed to diffuse following concentration gradients. For $NO_3^-$ and $NH_4^+$, strong chemical production happened in upper layers (913-771 hPa), where diffusion contributions at this altitude were negative, meaning the diffusion of new-generated $NO_3^-$ and $NH_4^+$ diffused through PBL. It is reasonable. However, $SO_4^{2-}$ diffusion were still positive at altitude where chemical production was strong, which seems against the concentration gradients. It might also be related to the constant $SO_4^{2-}$ profile in Fig. 8, which is interesting but I could not find clear explanations in this manuscript.

**Response:**
The vertical profile, chemical production, and diffusion of $SO_4^{2-}$ were different from those of $NO_3^-$ and $NH_4^+$. Because of the diffusion and chemical production of $SO_4^{2-}$, the profile of $SO_4^{2-}$ remained uniform and did not decrease with increasing altitude. To clarify that, we have added explanations in the second paragraph of Section 3.3.4: "Chem and Diff of $SO_4^{2-}$ were different from those of $NO_3^-$ and $NH_4^+$. For $SO_4^{2-}$, Chem was positive from the surface to 510 hPa with a peak around 819 hPa, and Diff was positive at 819-771 hPa but negative from 819 hPa to the surface, which resulted in the uniform $SO_4^{2-}$ profile."

5.  I suggest authors to summarize some highlights logistically in conclusions, e.g. what are the major differences in chemical mechanisms among $O_3$SPD, $PM_{2.5}$SPD and $O_3$&$PM_{2.5}$PD? What meteorological factors or synoptic patterns drives the differences? Also, although authors made very comprehensive analysis, one important question remained not very clear to me: Why $O_3$&$PM_{2.5}$PD only occurred at part of the $O_3$SPD or $PM_{2.5}$SPD? Which one among chemical mechanisms, vertical profile and meteorology drives the differences?

**Response:**

The major differences in chemical mechanisms among $O_3$SPD, $PM_{2.5}$SPD and $O_3$&$PM_{2.5}$PD were summarized by Figure 13 and described in the last paragraph of the conclusion section. We stated here that "Figure 13 summarizes the chemical and physical characteristics in $O_3$&$PM_{2.5}$PD, $O_3$SPD, and $PM_{2.5}$SPD in the BTH region. In $O_3$&$PM_{2.5}$PD, the strong chemical productions of $O_3$, $NO_3^-$, $NH_4^+$, and $SO_4^{2-}$ occur at high altitudes of 913-819 hPa where RH is high, and the accompanied downward airflow causes the stable concentrations at 944-819 hPa. The composited PBLH in $O_3$&$PM_{2.5}$PD is about 946.1 m, and the strong mixed diffusion underneath the PBLH leads to high concentrations of pollutants at the ground level. In contrast, $O_3$SPD occurs in hot and dry atmosphere with composited PBLH of 1073.5 m. Strong $O_3$ chemical production occurs around 819 hPa, and $O_3$ is then transported to the surface by downward air flow. The atmosphere is stable and stagnate when $PM_{2.5}$SPD occurs, with the lowest PBLH of 681.8 m. High RH (high chemical formation of $PM_{2.5}$) and the accumulation of aerosols lead to the highest surface-layer $PM_{2.5}$ in $PM_{2.5}$SPD."

As suggested, we have added the following sentences at the end of the last paragraph of the conclusion section to explain the driving factors of $O_3$&$PM_{2.5}$PD: "To summarize, $O_3$&$PM_{2.5}$PD were characterized by high $O_x$, SOR, and NOR, uniform vertical profiles at 975-819 hPa, which were caused by an anomalous high-pressure system at 500 hPa, strong southerlies and high RH at 850 hPa. Meteorological parameters around 850 hPa promoted strong chemical production of secondary aerosols and downward transport, resulting in the unique vertical profiles and high surface concentrations in $O_3$&$PM_{2.5}$PD.".

**Specific Comments:**
1.  Lines 48-49: Natural sources also have significant contributions to $PM_{2.5}$.
**Response:**
We have revised the sentence: "… and the major $PM_{2.5}$ components (nitrate ($NO_3^-$), ammonium ($NH_4^+$), sulfate ($SO_4^{2-}$), black carbon (BC), organic carbon (OC)) are mainly caused by anthropogenic emissions of aerosols and aerosol precursors.".

2.  Line 61: 'observations' should not be capitalized.
**Response:**
Revised.

3.  Lines 334-342: I suggest authors to add a table or figure in the main text or supplementary to show the OH evaluation.
**Response:**

We have added Table S1 in the Supplementary Material. The simulated OH concentrations agreed closely with the observed values.

**Table S1.** The comparisons of simulated OH concentrations with observations in BTH.

| Site | Time period | Observed OH | Simulated OH | Reference for observation |
|---|---|---|---|---|
| Beijing (39.6ºN, 116.2ºE) | Summer 2017 | $5.82 \times 10^6$ (averaged) | $2.4 \times 10^6$ (averaged) | Woodward et al. (2020) |
| Beijing (40ºN, 116.3ºE) | Winter 2017 | $1.5\text{-}2.0 \times 10^6$ (daytime) | $1.8\text{-}9.2 \times 10^5$ (averaged) | Ma et al. (2019) |
| Wangdu (38.7ºN, 115.2ºE) | Summer 2014 | $5\text{-}15 \times 10^6$ (daytime) | $3.7\text{-}9.5 \times 10^6$ (averaged) | Tan et al. (2017) |

4. Line 548: From the traditional synoptic definition, WPSH in eastern China should be regions with 500hPa geopotential height larger than 5880 m (or larger than 1520 m at 850hPa). I don't think the high pressure here is WPSH.

**Response:**
We have revised this sentence (see our response to your specific Comment #6).

5. Line 553: Northeast Cold Vortex is not necessary to abbreviate since it no longer appeared in the manuscript.

**Response:**
Revised.

6. Figure 12 and S10: I wonder could the synoptic patterns be clearer if using anomalies rather than absolute values?

**Response:**
Following the Reviewer's suggestion, we have added the anomalous synoptic patterns of Figure 12 and S10 (revised to Figure S12) as Figure S11 and S13, respectively, in the Supplementary Material. We have also added the following sentences in the second paragraph of Section 3.4: "the BTH region was controlled by westerlies and an anomalous high-pressure system at 500 hPa (Figure S11). At 850 hPa, BTH was at the west boundary of an anomalous anticyclone, and the associated strong anomalous southerlies at 850 hPa brought moist air to BTH (Fig. S12 and S13)"

[Figure]

**Figure S11.** Composites of the anomalies of winds (m s$^{-1}$) with SLP (sea level pressure) and with geopotential height at 850 hPa and 500 hPa for regional O$_3$SPD, PM$_{2.5}$SPD, and O$_3$&PM$_{2.5}$PD that were captured by the model during April-October of 2013-2020. The solid black rectangle indicates the BTH region.

[Figure]

**Figure S13.** Composites of the anomalies of winds (m s$^{-1}$) with RH (%) at 1000 hPa, 850 hPa, and 500 hPa for regional O$_3$SPD, PM$_{2.5}$SPD, and O$_3$&PM$_{2.5}$PD that were captured by the model during April-October of 2013-2020. The solid black rectangle indicates the BTH region.

**References:**

Barth, M.C., Rasch, P.J., Kiehl, J.T., Benkovitz, C.M., and Schwartz, S.E.: Sulfur chemistry in the national center for atmospheric research community climate model: description, evaluation, features and sensitivity to aqueous chemistry, J. Geophys. Res. Atmos., 105, 1387–1415, 2000.

Dovrou, E., Rivera-Rios, J.C., Bates, K.H., and Keutsch, F.N.: Sulfate formation via cloud processing from isoprene hydroxyl hydroperoxides (ISOPOOH), Environ. Sci. Technol., 53, 12476–12484, 2019.

Duan, F., He, K., Ma, Y., Yang, F., Yu, X., Cadle, S. H., Chan, T., and Mulawa, P. A.: Concentration and chemical characteristics of PM2.5 in Beijing, China: 2001–2002, Sci. Total Environ., 355(1–3), 264–275, https://doi:10.1016/j.scitotenv.2005.03.001, 2006.

Ervens, B.: Modeling the processing of aerosol and trace gases in clouds and fogs, Chem. Rev., 115 (10), 4157–4198, 2015.

Ervens, B.: Progress and problems in modeling chemical processing in cloud droplets and wet aerosol particles, ACS Symp. Ser., 1299, 327–345, 2018.

Harris, E., Sinha, B., van Pinxteren, D., Tilgner, A., Fomba, K.W., Schneider, J., Roth, A., Gnauk, T., Fahlbusch, B., Mertes, S., Lee, T., Collett, J., Foley, S., Borrmann, S., Hoppe, P., and Herrmann, H.: Enhanced role of transition metal ion catalysis during in-cloud oxidation of $SO_2$, Science, 340, 727–730, 2013.

Liao, H., Adams, P. J., Chung, S. H., Seinfeld, H. J., Mickley, L. J., and Jacob, D. J.: Interactions between tropospheric chemistry and aerosols in a unified general circulation model, J. Geophys. Res., 108(D1), 4001, https://doi:10.1029/2001JD001260, 2003.

Ma, X. F., Tan, Z. F., Lu, K. D., Yang, X. P., Liu, Y. H., Li, S. L., Li, X., Chen, S. Y., Novelli, A., Cho, C. M., Zeng, L. M., Wahner, A., and Zhang, Y. H.: Winter photochemistry in Beijing: Observation and model simulation of OH and $HO_2$ radicals at an urban site, Sci. Total Environ., 685, 85−95, 2019.

Ma, X.Y., and Salzen, K.V.: Dynamics of the sulphate aerosol size distribution on a global scale, J. Geophys. Res. Atmos., 111 (D8), 2006.

Mu, Q., and Liao, H.: Simulation of the interannual variations of aerosols in China: Role of variations in meteorological parameters, Atmos. Chem. Phys., 14, 9597-9612, https://doi:10.5194/acp-14-9597-2014, 2014.

Tan, Z. F., Fuchs, H., and Lu, K. D.: Radical chemistry at a rural site (Wangdu) in the North China Plain: Observation and model calculations of OH, HO2 and RO2 radicals, Atmos. Chem. Phys., 17(1): 663−690, 2017.

Woodward-Massey, R., Slater, E. J., Alen, J.: Implementation of a chemical background method for atmospheric OH measurements by laser-induced fluorescence: characterisation and observations from the UK and China, Atmos. Meas. Tech., 13(6): 3119−3146, 2020.

Xuan, J., Liu, G., and Du, K.: Dust emission inventory in northern China, Atmos. Environ., 34(26), 4565–4570, https://doi:10.1016/S1352-2310(00) 00203-X, 2000.

Ye, B., Ji, X., Yang, H., Yao, X., Chan, C. K., Cadle, S. H., Chan, T., and Mulawa, P. A.: Concentration and chemical composition of PM2.5 in Shanghai for a 1-year period, Atmos. Environ., 37(4), 499–510, https://doi:10.1016/S1352-2310(02)00918-4, 2003.

Zhao, X. J., Zhao, P. S., Xu, J., Meng, W., Pu, W. W., Dong, F., He, D., and Shi, Q. F.: Analysis of a winter regional haze event and its formation mechanism in the North China Plain, Atmos. Chem. Phys., 13, 5685–5696, https://doi:10.5194/acp-13-5685-2013, 2013.